# Projected climate change in Fennoscandia — and its relation to ensemble spread and global trends

Gustav Strandberg[1,2], August Thomasson[1,3], Lars Bärring[1], Erik Kjellström[1], Michael Sahlin[1], Renate Anna Irma Wilcke[1], Grigory Nikulin[1]

[1] Rossby Centre, Swedish Meteorological and Hydrological Institute, Norrköping, SE-601 76, Sweden
[2] Centre for Climate Science and Policy Research, Linköping University, Linköping SE-581 83, Sweden
[3] Department of Physics, Lund University, Lund, SE-223 64, Sweden

*Correspondence to*: Gustav Strandberg (gustav.strandberg@smhi.se)

**Abstract.** The need for information about climate change is great. This information is usually based on climate model data
— data that often have systematic biases. Furthermore, climate information is based on ensembles of climate model which raises the question about how such ensembles are affected by the choice of models and emission scenarios. Here, we aim to give a description of climate change in Sweden and neighbouring countries, as well as a discussion on how local climate change relates to global warming. We present climate change projections based on bias adjusted Euro-CORDEX (Coordinated Regional Downscaling Experiment) regional climate model data centred over Sweden. Global warming results
in higher temperature, more warm days and fewer cold days in Sweden. The regional climate models replicate the signal of the driving global models. Yet, the model spread is smaller than in the full CMIP5 ensemble, which means that the RCMs do not fully represent the potential model spread. The choice of emission scenario has minimal effect on the calculation of mean climate change at a global warming level of 2 degrees. This implies that it would be safe to mix emission scenarios in calculations of global warming levels, at least up to +2°C, and as long as mean values are concerned. Moreover, the
differences in local and global warming rates seem to decrease with time, suggesting that climate change in Sweden may currently be at its fastest.

## 1 Introduction

Unless strong reductions in emissions of greenhouse gases are implemented, global warming will within the 21st century likely reach +2 °C compared to pre-industrial values (Forster et al., 2024). The temperature response in Europe is correlated
to, but warms at stronger rates, than the global temperature change (IPCC, 2021). Since 1850-1900 the global temperature-increase of 1.3°C is translated to a warming of 2.3°C in Europe and 3.3°C in the Arctic (C3S, 2024).

The current strong global warming calls for climate adaptation in all parts of society. Adaptation measures must be based on informed decisions to be cost efficient and to avoid maladaptation (IPCC, 2022). Thus, there is a great need for climate data to support decision making and adaptation.

A way to avoid the discussion on which emission scenario to use and which scenario is the most likely — a discussion that is sometimes heated (Hausfather & Peters, 2020; Schwalm et al., 2020) — is to use global warming levels (GWL). Instead of a fixed period of time in a certain scenario GWLs focus on the period when a particular level of global warming is reached. For example, GWL2 is the period when +2°C global warming is reached compared to pre-industrial times. This period may occur at different times in different models – instead of consistency in time between the members of the ensemble there is

thus a consistency in the magnitude of temperature increase. In that way, using GWLs is a powerful method since it is possible to mix simulations using different scenarios to create larger ensembles; and since it reduces the uncertainty around the choice of emission scenarios (Maule et al., 2017). One example of how to use GWLs for regional data is found in Strandberg et al. (2024b). The mixing of emission scenarios in GWLs can nevertheless be criticised because the trends are different between scenarios (Bärring & Strandberg, 2018); a GWL based on RCP2.6 does not have the same characteristics

as a GWL based on RCP8.5. This means that also a GWL ensemble is sensitive to how it is constructed with regards to which models and scenarios that are used as input. We want to investigate the robustness of the ensembles and how the simulated climate at a specific GWL is affected by the choice of emissions scenario, and global and regional models.

Climate models are our main tool to make projections of future climate change. Climate modelling is computationally expensive which means that global climate models (GCMs) are usually run on relatively coarse horizontal resolutions

(typically 100 - 300 km). On the other hand, regional climate models (RCMs) can be run at higher resolution (typically 5-20 km) since they cover a smaller part of the globe. Therefore, RCMs can provide new information despite being governed by the driving GCM (e.g. Vautard et al., 2020; Strandberg & Lind, 2020). Topographical features, such as coastlines or mountains, are better described with higher model resolution. Furthermore, RCM simulations give more details and a better representation of physical processes, especially local events like convective rain and short-duration extreme events (e.g.

Olsson et al., 2015; Prein et al., 2015; Rummukainen, 2016; Lind et al., 2020).

CORDEX (Coordinated Regional climate Downscaling Experiment, Jacob et al., 2024) provides the most comprehensive RCM ensemble for Europe on a high resolution. A key advantage of using climate model ensembles, like the CORDEX ensemble, is that they allow for a probabilistic assessment of potential changes, uncertainty estimations and a wider set of statistical tests (Déqué et al., 2012; Coppola et al., 2021). By relying on only one or very few model simulations there is a

risk that you only sample a small part of the possible outcomes. Furthermore, one simulation is not enough to estimate model sensitivity to emissions of greenhouse gases, model uncertainty, or natural variability (e.g. von Trentini et al., 2019; Christensen and Kjellström, 2020; 2021).

Since all parts of society are affected by climate change, it is crucial to have a well-founded description of it—especially

given the significant economic investments that will be based on climate projections. By 'a good description,' we mean an ensemble that is both accurate and representative, and, not least, large enough to enable the assessment of the significance and robustness of simulated climate change. Additionally, a general understanding of ensembles is necessary. It is important to know how an ensemble's characteristics is shaped by the models and scenarios that compose it.

Here we present a new dynamically downscaled and bias adjusted ensemble of climate projections for Sweden. Compared to the previous ensemble (Kjellström et al., 2016) improvements include higher horizontal resolution in the RCMs, bias adjustment of the results, more ensemble members and more indicators developed in dialogue with users to meet their needs. Climate model projections is an important tool for illustrating various aspects of climate change and how it could impact society. This data is used to support decision makers' work on climate change adaptation in Sweden. Rather than relying solely on standard climatological variables, inclusion of climate indicators in the assessment enables insights regarding impacts that are more directly relevant to society. These indicators should support climate adaptation, by serving as decision support and informing the general public.

Since these data cover Fennoscandia and the Baltic States, they may also be applicable to surrounding countries. They are based on RCP (Representative Concentration Pathways) scenarios and CMIP5 (Coupled Model Intercomparison Project Phase 5; Taylor et al., 2022) global models. The Swedish climate service (SMHI, 2025) relies on these data, and at least until a CMIP6-based downscaled ensemble becomes available, they will continue to be used. Since this RCM ensemble is already existing and used it is important to also discuss how the ensemble is constructed and how that influence the characteristics of the ensemble.

The RCM ensemble presented here is already existing and used. Therefore, it is important to also discuss how the ensemble is constructed and how that influence the characteristics of the ensemble; as a service to all users. This study aims at four general topics:

    i)        Projected climate change in Fennoscandia. This paper serves as a general overview of projected climate change in Sweden based on the best available material, making this the currently most comprehensive projection of climate change in the region and a basis for further research and decision-making.

    ii)       How local trends in climate relate to global warming. Fennoscandia is known to have a warming trend that greatly exceeds the global trend, but still with a relatively linear relationship (C3S, 2024). It is, however, unknown if this relationship will persist in the future.

    iii)     Model spread in the RCM ensemble compared to the spread of the larger CMIP5 ensemble. Since the RCM ensemble is forced by a sub-set of available GCMs the model spread is potentially reduced. This would mean that information is lost in the RCM ensemble.

    iv)     Since it is likely that global warming will reach +2 °C within this century, and since the Paris Agreement (UNFCCC, 2015) speaks of keeping the temperature rise to well below 2 °C, it is natural that descriptions of projected climate change are formed around a two degree warmer world. The question is how such 'global warming levels' are influenced by the climate models and emission scenarios used to calculate them.

## 2 Methods

### 2.1 The Euro-CORDEX ensemble

The presented data describing simulated present and future climates are based on the Euro-CORDEX ensemble covering Europe with a grid spacing of 0.11°, which approximately equals 12.5×12.5 km (Jacob et al., 2014). Within CORDEX several global climate models (GCMs) are used to force a number of regional climate models (RCMs). Every six hours the RCMs read data from the GCMs on the boundary of their model domains. The boundary conditions include temperature, pressure, humidity and wind at several vertical levels as well as sea surface temperature and sea ice conditions.

The Euro-CORDEX RCMs used here are forced by a subset of GCMs from the CMIP5. The RCMs were evaluated using observations and were judged to generally perform well in the historical climate of the late 20th century (Vautard et al., 2021). This does not mean that the CORDEX simulations are without systematic errors. Vautard et al (2021) conclude that the simulations are generally too wet, too cold and too windy compared to observations. Some of the discrepancies between GCMs and RCMs, as well as the weak warming trend, could be explained by a too simple description of aerosol forcing (Boé et al., 2021; Katragkou et al., 2024).  Projections for the 21st century from the RCMs have previously been assessed for Europe by Coppola et al. (2021). The simulations and their combinations of GCMs, RCMs and RCPs are listed in Table 1. As this study is based on an already existing ensemble that is already being used (SMHI, 2025), we have not made any choices of excluding simulations. To add or exclude members would mean that we investigate another ensemble than the one used in for example the SMHI climate service. When the ensemble was created it was created after a 'the more the better'-approach, which means that as many simulations as possible are used.

**Table 1 The simulations used in this study and the GCMs, RCMs and RCPs that they consist of. Members that are part of an ensemble consistent across all RCPs (RCM17) are marked with an '*'.**

| Driving GCM | | RCM | Scenario | | | |
|---|---|---|---|---|---|---|
| GCM | No. | | RCP2.6 | RCP4.5 | RCP8.5 | |
| CNRM-CERFACS-CNRM-CM5 | r1i1p1 | CLMcom-ETH-COSMO-crCLIM-v1-1 | | | X | |
| | | CNRM-ALADIN63 | x | x | X | * |
| | | DMI-HIRHAM5 | | | X | |
| | | GERICS-REMO2015 | X | | X | |
| | | IPSL-WRF381P | | | X | |
| | | KNMI-RACMO22E | X | X | X | * |
| ICHEC-EC-EARTH | r1i1p1 | CLMcom-ETH-COSMO-crCLIM-v1-1 | | | X | |
| | | DMI-HIRHAM5 | | | X | |

| GCM | Ensemble | RCM | | | | |
|---|---|---|---|---|---|---|
| | | KNMI-RACMO22E | | X | X | |
| | | SMHI-RCA4 | | | X | |
| | r3i1p1 | CLMcom-ETH-COSMO-crCLIM-v1-1 | | | X | |
| | | DMI-HIRHAM5 | X | X | X | * |
| | | KNMI-RACMO22E | | | X | |
| | | SMHI-RCA4 | | | X | |
| | r12i1p1 | CLMcom-CCLM4-8-17 | X | X | X | * |
| | | CLMcom-ETH-COSMO-crCLIM-v1-1 | | | X | |
| | | DMI-HIRHAM5 | | | X | |
| | | ICTP-RegCM4-6 | | | X | |
| | | GERICS-REMO2015 | X | X | X | * |
| | | KNMI-RACMO22E | X | X | X | * |
| | | MOHC-HadREM3-GA7-05 | X | | X | |
| | | SMHI-RCA | X | X | X | * |
| | | IPSL-WRF381P | | | X | |
| IPSL-IPSL-CM5A-MR | r1i1p1 | DMI-HIRHAM5 | | | X | |
| | | GERICS-REMO2015 | | | X | |
| | | KNMI-RACMO22E | | | X | |
| | | SMHI-RCA4 | | X | X | |
| | | IPSL-INERIS-WRF331P | | X | X | |
| MIROC-MIROC5 | r1i1p1 | CLMcom-CCLM4-8-17 | X | | X | |
| | | GERICS-REMO2015 | X | | X | |
| MOHC-HadGEM2-ES | r1i1p1 | CLMcom-CCLM4-8-17 | | X | X | |
| | | CLMcom-ETH-COSMO-crCLIM | | | X | |
| | | CNRM-ALADIN63 | | | X | |
| | | DMI-HIRHAM5 | X | X | X | * |
| | | GERICS-REMO2015 | X | X | X | * |

| | | | | | | |
|---|---|---|---|---|---|---|
| | | ICTP-RegCM4-6 | X | | X | |
| | | KNMI-RACMO22E | X | X | X | * |
| | | MOHC-HadREM3-GA7-05 | X | | X | |
| | | SMHI-RCA4 | X | X | X | * |
| | | IPSL-WRF381P | | | X | |
| MPI-M-MPI-ESM-LR | r1i1p1 | CLMcom-CCLM4-8-17 | X | X | X | * |
| | | CLMcom-ETH-COSMO-crCLIM-v1-1 | | | X | |
| | | CNRM-ALADIN63 | | | X | |
| | | DMI-HIRHAM5 | | | X | |
| | | MPI-CSC-REMO2009 | X | X | X | * |
| | | ICTP-RegCM4-6 | X | | X | |
| | | KNMI-RACMO22E | X | | X | |
| | | MOHC-HadREM3-GA7-05 | | | X | |
| | | SMHI-RCA4 | X | X | X | * |
| | | IPSL-WRF381P | | | X | |
| | r2i1p1 | CLMcom-ETH-COSMO-crCLIM-v1-1 | | | X | |
| | | MPI-CSC-REMO2009 | X | X | X | * |
| | | SMHI-RCA4 | | | X | |
| NCC-NorESM1-M | r1i1p1 | CLMcom-ETH-COSMO-crCLIM-v1-1 | | | X | |
| | | CNRM-ALADIN63 | | | X | |
| | | DMI-HIRHAM5 | | X | X | |
| | | GERICS-REMO2015 | X | X | X | * |
| | | ICTP-RegCM4-6 | X | | X | |
| | | KNMI-RACMO22E | X | | X | |
| | | MOHC-HadREM3-GA7-05 | | | X | |
| | | SMHI-RCA4 | X | X | X | * |
| | | IPSL-WRF381P | | | X | |

115

## 2.2 Bias adjustment

To minimise systematic errors the Euro-CORDEX ensemble was bias adjusted using the method "Multi-scale Bias Adjustment" available in MIdAS (Berg et al., 2022). MIdAS is based on quantile mapping 'day-of-year' adjustments (Themeßl et al 2011; Wilcke et al., 2013). This means that the distribution used to adjust the data is different for each day of the year. MIdAS is aiming at preserving the trend in future projections and does perform similar to methods that explicitly preserve trends (Berg et al., 2022). As reference data the SMHI gridded climatology (SMHIGridClim) data set (Andersson et al., 2021) was used. SMHIGridClim covers Fennoscandia and the Baltic states (region A in Fig 1), which means that the bias adjusted ensemble covers a smaller domain centred over Sweden, instead of the entire European domain. The bias adjustment was made using the period 1980-2000 as a reference. The variables tas, tasmin, tasmax and pr (see Table 2 for explanations) were adjusted in all gridpoints within the domain. Below, any further mentions of the *CORDEX RCMs* refers to this bias adjusted ensemble covering Fennoscandia and the Baltic states (region A in Fig 1).

## 2.3 Calculation of indicators

To assess climate change, a set of climate indicators are calculated using the software package Climix (Bärring et al., 2024). A number of indicators were identified, building on the work of Kjellström et al (2016), and together with the Swedish County Administrative Boards and other governmental agencies, that can describe relevant changes in climate. The indicators are meant to be relevant for large parts of society, but agriculture (Strandberg et al., 2024a) and the energy sector (Strandberg et al., 2024b) have also been specifically targeted. The indicators used in this study are listed in Table 2. The indicators are presented as averages for the 30-year periods used in the SMHI web service (SMHI, 2025): the reference period 1971-2000 and the future periods 2011-2040, 2041-2070 and 2071-2100. WMO recommends 1961-1990 as reference period for descriptions of climate change (WMO, 2017), but since several RCM simulations start 1971, a compromise is to use 1971-2000.

The GWLs are calculated for each driving GCM and based on the global mean surface temperature (GMST) using the period 1850-1900 as a reference, following the protocol in the IPCC-WG1 Atlas (Iturbide et al., 2022). A GWL is reached when the GMST for a moving 20-year time window for the first time passes that level. For example: GWL2 occurs when the GMST for the first time is 2°C more than in the reference period. The timing of a GWL is represented by a central year. In this study we use 30-year periods for each GWL stretching from 15 years before the central year to 14 years after.

**Table 2 Definitions and short names of indicators.**

| Indicator | Name | Definition | Unit |
|---|---|---|---|
| Average temperature | tas | The daily average temperatures | °C |
| Minimum temperature | tasmin | The daily minimum temperatures averaged over a selected period | °C |

| Maximum temperature | tasmax | The daily maximum temperatures averaged over a selected period | °C |
|---|---|---|---|
| Frost days | fd | Number of days with daily minimum temperature < 0°C | days |
| Summer days | su | Number of days with daily maximum temperature above 20°C | days |
| Consecutive summer days | csu | Longest period with consecutive days with daily maximum temperature above 20°C | days |
| Days with zero crossings | nzero | Number of days over with daily maximum temperature above 0°C and daily minimum temperature below 0°C | days |
| Precipitation | pr | Average precipitation amount | mm/ mon |
| Days with heavy precipitation | r10mm | Number of days with precipitation amount of more than 10 mm | days |
| Dry days | dd | Number of days with precipitation less than 1 mm | days |

**2.4 GCM ensembles**

The bias adjusted CORDEX RCMs are compared to two GCM ensembles.

*CORDEX GCMs*: consisting of the GCMs actually used to drive the RCMs (leftmost column in Table 1) (ensemble sizes 5, 9 and 9 for scenarios RCP2.6, RCP4.5 and RCP8.5 respectively). This ensemble includes several realisations for some GCMs since they are used to force RCMs.

*CMIP5 GCMs*: consisting of all CMIP5 models available on the Earth System Grid Federation, but restricted to one realisation per GCM to avoid overweight on certain GCMs (ensemble sizes 24, 28 and 34 for scenarios RCP2.6, RCP4.5 and RCP8.5 respectively)

The GCMs are not bias adjusted. For all GCMs the grid points falling within the Fennoscandian region (A in Figure 1) are 155 used to calculate ensemble mean and spread for the region. For both GCM ensembles the GMST as 30-year averages for the reference period 1971-2000 and the future periods 2011-2040, 2041-2070 and 2071-2100 are calculated.

**2.5 Selection and analyses of sub-ensembles**

We want to study the relative importance of the choice of RCP, GCM and RCM at a specific GWL. To get a consistent ensemble across RCPs we select only the combinations of GCMs and RCMs that simulated all of the scenarios RCP2.6, RCP4.5 and RCP8.5 (indicated with '*' in Table 1). From these 17 combinations of GCMs, RCMs and RCPs (i.e. 51 RCM simulations) we construct sub-ensembles where all 17 members use the same RCP, GCM or RCM. We call this ensemble *RCM17*. If we would use the full CORDEX RCM ensemble it would be difficult to separate the effect of different ensemble

sizes and the effect of models or scenarios. The RCM17 ensemble is used in section 3.5.

To illustrate the procedure, we make the hypothetical case of three GCMs (GCM1-3) and three RCMs (RCM1-3) combined in different ways (Table 3). A sub-ensemble only using GCM1 would include all RCMs forced by GCM1, i.e. the simulations in row R1 in Table 3, i.e.three simulations. In the same way the sub-ensemble based on GCM2 consist of two

simulations. Sub-ensembles using only one RCM use all simulations with one RCM forced by different GCMs, i.e. one of the columns C1-3. The sub-ensemble based on RCM1 has three simulations. Sub-ensembles based on one scenario use all simulations run with that scenario.

**Table 3 Hypothetical sketch of how three GCMs (GCM1-3) could be downscaled by three RCMs (RCM1-3) and how the sub-ensemble strategy works**

|    |      | C1 RCM1 | C2 RCM2 | C3 RCM3 |
|----|------|---------|---------|---------|
| R1 | GCM1 | X       | X       | X       |
| R2 | GCM2 | X       |         | X       |
| R3 | GCM3 | X       |         |         |


The GWLs are calculated based on the GMST using the period 1850-1900 as a reference, following the protocol in the IPCC-WG1 Atlas (Iturbide et al., 2022). A GWL is reached when the GMST for a moving 20-year time window for the first time passes that level. For example: GWL2 occurs when GMST for the first time is 2°C more than in the reference period.

The timing of a GWL is represented by a central year. In this study we use 30-year periods for each GWL stretching from 15 years before the central year to 14 years after.We analysed GWL1.5 and GWL2. GWL1.5 is reached in all scenarios, while GWL2 is reached in RCP4.5 and RCP8.5, but not in RCP2.6. Already at GWL3 most of the RCP4.5 are discarded since they do neat reach that level of warming. The lack of different scenarios and the smaller ensemble size makes GWL3 a less interesting, and less useful, case for this analysis.

Seven GCMs are in different ways combined with seven different RCMs. This means that we have 2 RCP-based, 7 GCM-based and 7 RCM-based sub-ensembles. In order to determine if any of the model combinations is significantly different from the others we perform two statistical tests, with the null hypothesis that any given two ensembles have the same average. We use a one-way ANOVA (Analysis of variance) (Press, 1972) test, which tests whether two or more groups have

the same average or not. If the number of groups is equal to 2, which is the case when we compare RCP-based ensembles, a one-way ANOVA is the same as a student's t-test. The ANOVA test does not specify which sub-ensemble is different from the others, if any. To find the different sub-ensemble(s) a post-hoc test is used, the so-called Tukey's honestly significant difference (Tukey, 1949). It is performed after a successful ANOVA test and compares all sub-ensembles pair-wise with a studentised q distribution. The tests are done for a region in northern Sweden and a region in southern Sweden representing different climatic conditions in Sweden (regions C and D in figure 1). A significance level with a family-wise error of 95 % is used. This means that the probability of one or more false positives among all grid points cells is 5 % instead of a 5 % false positive rate in each individual grid point, if no correction is applied. The analyses were made for tas, csu, tasmin, tasmax, pr, dd, cdd, su, r20mm and nzero.

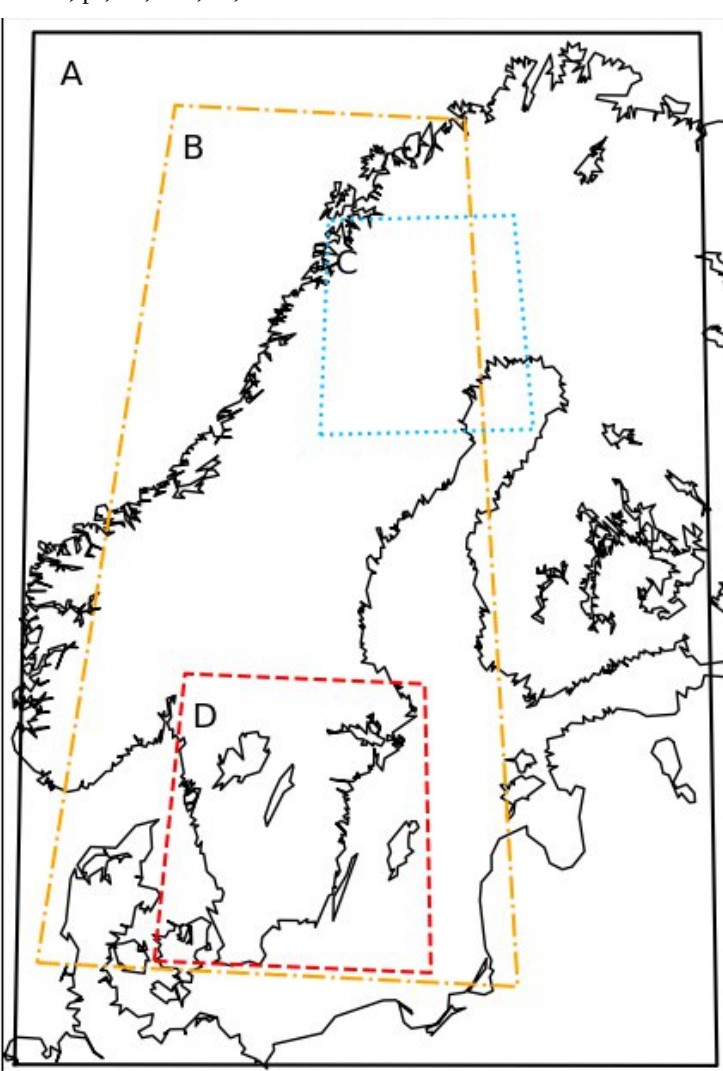

**Fig 1: Maps of regions used in analyses in this paper. A) Fennoscandian region (black, full line) is the domain on which bias adjustment is applied, B) Scandinavia (dash dotted orange line), C) northern Sweden (dotted blue line), D) southern Sweden (dashed red line).**

## 3 Results and discussion

Here, we start by describing average climate changes according to the CORDEX RCM ensemble. To understand these trends, they are then put in relation to the trend in GMST in the driving GCMs (CORDEX GCMs). This is followed by a comparison in ensemble spread between CORDEX RCMs, CORDEX GCMs and a larger ensemble of CMIP5 GCMs to see how much of the potential spread that is lost by not using all available GCMs. Section 3 is concluded by an investigation of how the description of a GWL based on the RCM17 ensemble is influenced by the GCMs, RCMs and RCPs of which it is constructed.

### 3.1 Projected change in temperature and temperature-based indicators

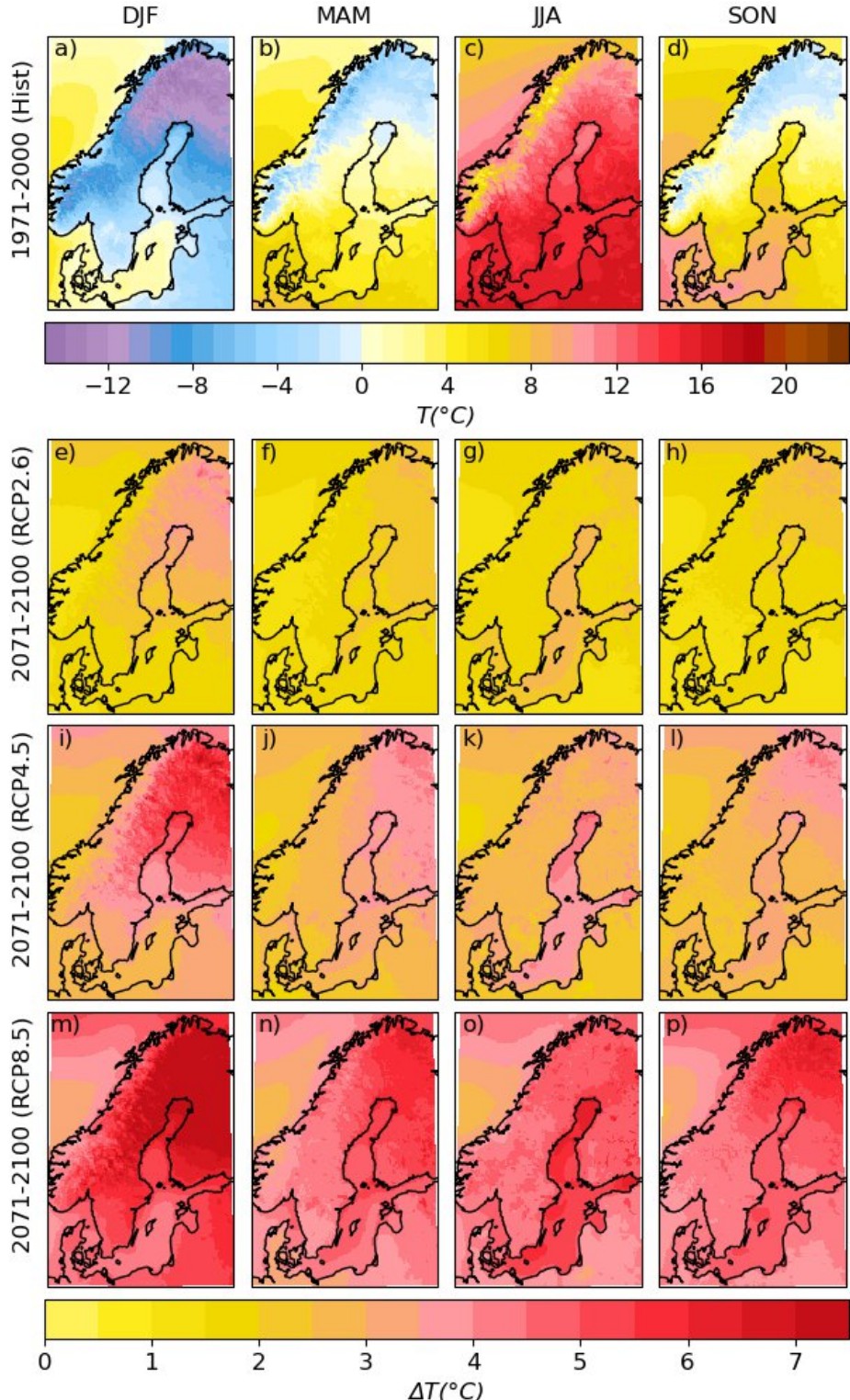

**Figure 2: Ensemble mean temperatures of the CORDEX RCMs (°C) in winter (DJF, first column), spring (MAM, second column), summer (JJA, third column) and autumn (SON, fourth column). First row shows absolute values for 1971-2000. Rows 2 to 4 show anomalies from 1971-2000 to 2071-2100 according to scenarios RCP2.6, RCP4.5 and RCP8.5, respectively.**

The mean temperatures (tas) are projected to increase in all seasons and in all emission scenarios across the domain (Fig. 2). The increase in annual mean temperature in Sweden to the end of the century is 1-2°C in RCP2.6, 2-4°C in RCP4.5 and 4-6°C in RCP8.5, with larger differences in the north than in the south. The changes scale in such a way that RCP8.5 in the close future shows similar warming to RCP2.6 in the middle of the century, and RCP8.5 in the middle of the century is similar to RCP4.5 in the end of the century (Strandberg et al., 2024a).

In Fennoscandia, we highlight two climate change patterns for temperature: winter is the season with the fastest warming rate, and the northern parts of the region is warming faster than the southern parts. In RCP2.6 the warming in winter is 1.5-3.5°C from south to north (Fig 2e) and 1.5-2°C in summer (Fig 2g). In RCP8.5, the corresponding numbers are 4.5-8°C in winter (Fig 2m), and 4-5°C in summer (Fig 2o). This means that the warming is larger in winter, but also the difference between north and south.

The temperature change is especially large for the daily minimum temperature (tasmin) (Fig 3a). For example, in RCP4.5 the increase in tasmin is 3-6.5°C, to be compared to an increase in annual tas of 2-4°C. The increase in daily maximum temperature (tasmax) is comparable to tas, 2-3.5°C (Fig 3b). A warmer climate means fewer cold days and more warm days. Accordingly, the number of frost days (fd) is projected to decrease, though relatively uniform across the domain (Fig 3c). RCP4.5 gives a reduction of 40-50 days in most of Fennoscandia and the Baltic countries. The change is somewhat smaller in parts of the Scandinavian mountain chain (decrease in fd with 30-40 days), and larger over the Bothnian Sea and Bothnian Bay (reduction of 65 days or more). See figure S1 for absolute values of the indicators in 1971-2000 and figures S2-4 for climate anomalies in all scenarios RCP2.6, RCP4.5 and RCP8.5.

The increase in the number of summer days (su) according to RCP4.5 stretches from zero, or just a few days, in large parts of the mountain chain and over most of the sea in the domain to 20-24 days in southern Sweden and Denmark (Fig 3d). The number of days with zero crossings (nzero) shows a general decrease on the annual scale (Fig 3e). In winter, however, nzero increases in most of the domain, except for Denmark, southern Sweden and the Baltic countries (Fig S5). In these areas the temperatures will not drop below zero degrees as often, whereas in parts of northern Sweden the increase is as much as around 10 days (roughly corresponding to an increase of 50 %).

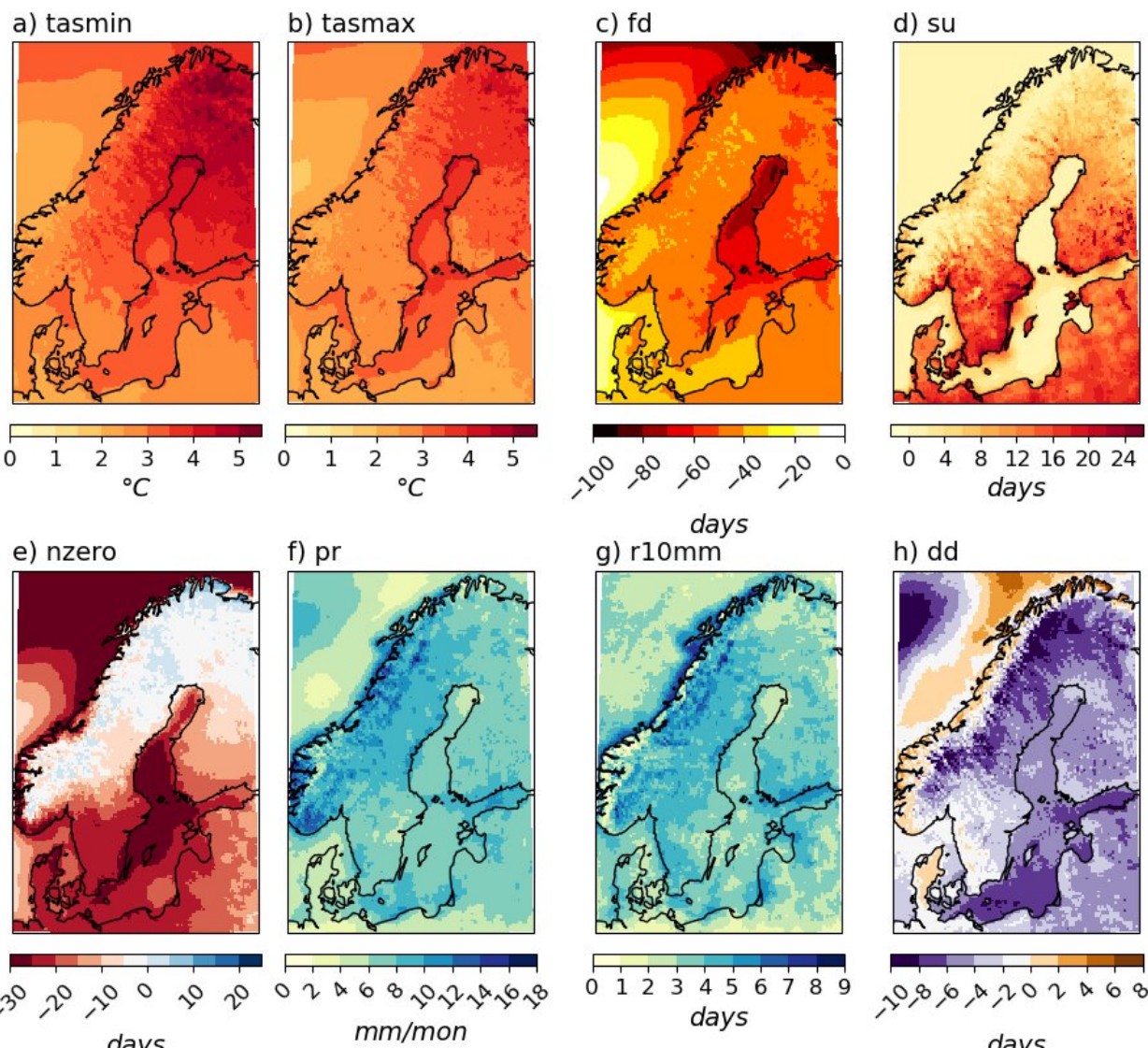

**Figure 3: Annual climate change anomalies in the CORDEX RCMs between 1971-2000 and 2071-2100 according to scenario RCP4.5. The maps show ensemble means of a) daily minimum temperature (tasmin, °C), b) daily maximum temperature (tasmax, °C), c) number of frost days (fd, days), d) number of summer days (su, days), e) number of days with zero crossings (nzero, days), f) mean precipitation (pr, mm mon-1), g) number of days with heavy precipitation (r10mm, days) and h) dry days (dd, days). See table 1 for definitions of the indicators.**

## 3.2 Projected change in precipitation and precipitation-based indicators

The annual average precipitation shows a general increase in the future (Fig 3f). According to RCP4.5 the increase in annual average daily precipitation is 5-10 mm mon-1 in large parts of the domain, the increase along the Norwegian west coast is up

to 15 mm mon$^{-1}$. In RCP2.6 the increase is smaller, 2-6 mm mon$^{-1}$ (fig. S2), and in RCP8.5 larger, 8-15 mm mon$^{-1}$ (fig S4). For most of the domain, the increase is larger in winter and smaller in summer compared to the annual change (Figs S5-S8). Denmark and southern Sweden show changes in summer precipitation close to zero. On the annual scale all models agree on the sign of change in most of the domain and all RCPs (Fig 3., Figs S2-S4). The signal is least robust in RCP2.6 since the change is smaller there, and since precipitation has generally large variability. The number of days with heavy precipitation (r10mm) is projected to increase with 3-5 in most of the domain (to be compared with 10-12 days in the reference period) (Fig 3g). The change is smaller in RCP2.6 (up to +2 days increase) and larger in RCP8.5 (+4-8 days) (Figs S1g & S2g). The number of dry days is projected to decrease with 1-8 days (Fig 3h). The signal is not robust, half of the ensemble members give increasing number of dry days, and half of the members decreasing.

## 3.3 Local trends in climate indicators related to global warming

Climate change is unevenly distributed across the globe. In Scandinavia, like most of Europe, the overall warming since pre-industrial times was about twice the global mean at the end of the 20th century (Schimanke et al., 2022; WMO, 2023). In this section, we take a look at how specific features of local climate change in the CORDEX RCMs relates to the change in global mean surface temperature (GMST) in the CMIP5 GCMs (fig 4).

The almost two-to-one relationship between global and local temperature is seen for mean, minimum and maximum temperatures in the early parts of the 21st century until the period 2011-2040 (Fig. 4 a-c). Within this period the ratio between regional and global warming is 1.6-1.8. With increasing global warming this relationship weakens and approaches a one-to-one relationship between change in global and local temperatures (i.e. parallel to the dotted lines in Figs 4a-c). In RCP4.5 and RCP8.5 the trend from 2041-2070 to 2071-2100 is roughly one to one (1.1-1.2), suggesting that the faster warming in Scandinavia will slow down as GMST increases. A conclusion of this could be that the ratio between warming in Scandinavia and global warming is at its largest in the beginning of the 21st century.

For indicators representing cold conditions the trend gets flatter in RCP8.5 reflecting that the potential for change decreases. For example: the number of frost days cannot be less than zero. For warm indicators, the trend instead gets steeper. The number of summer days is based on a temperature threshold, which means that there is a sudden effect when temperatures exceed the threshold. Consequently, the increase may be limited if the number of days above the threshold is already large. Indicators for precipitation shows continued increase under global warming. Here, results both for pr and r10mm show a slightly weaker trend in RCP8.5 than in the other two scenarios.

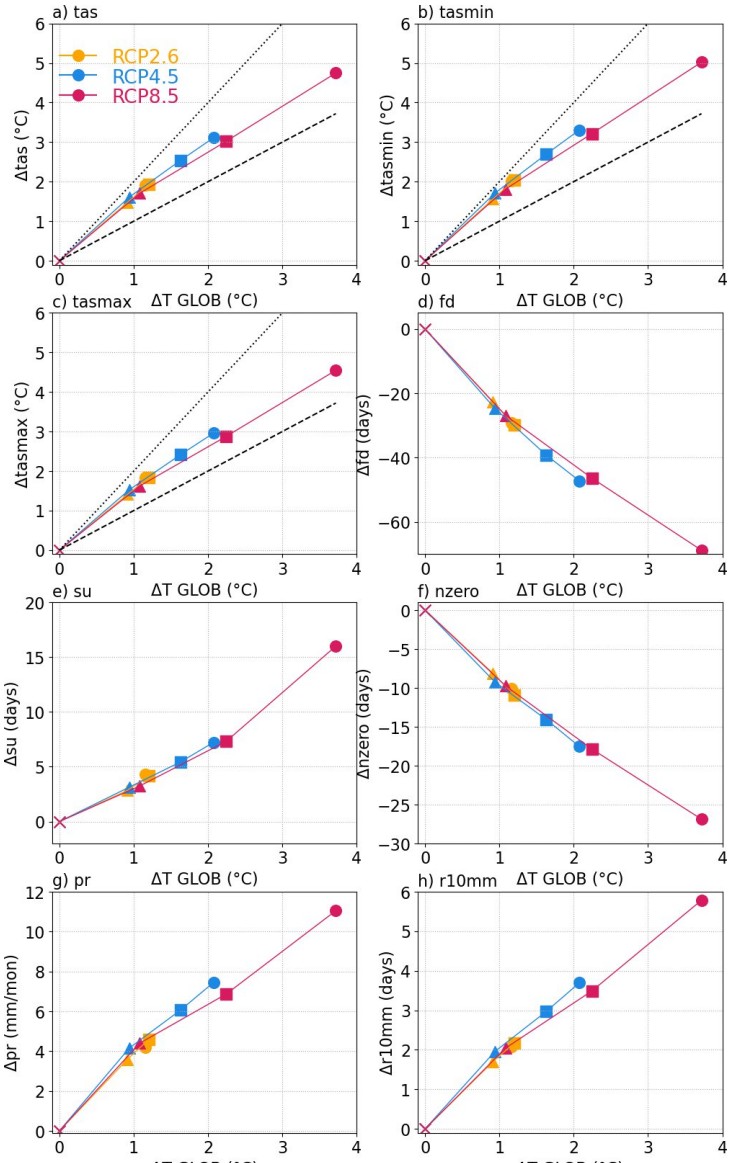

Fig 4: Changes relative to 1971-200 for the Fennoscandian domain (region A in Fig 1) in the CORDEX RCMs (y-axes) against that in global annual temperature in the driving CORDEX GCMs (x-axes), relative to the period 1971 — 2000. Different indicators are calculated based on RCM data: a) mean temperature (tas, °C), b) minimum temperature (tasmin, °C), c) maximum temperature (tasmax, °C), d) no. of frost days (fd, days), e) no. of summer days (su, days), f) no. of days with zero crossings (nzero, days) g) precipitation (pr, mm mon-1) h) no. of days with heavy precipitation (r10mm, days). Markers represent the periods 1971 — 2000 (cross), 2011 — 2040 (triangle), 2041 — 2070 (square), 2071 — 2100 (circle) for emissions scenarios RCP2.6 (green), RCP4.5 (orange) and RCP8.5 (light blue). In panels a-c the one-to-one relationship is shown with a dashed line, and the two-to-one with a dotted line.

**3.4 Model spread in the CORDEX RCM and CORDEX GCM ensembles compared to the spread in the CMIP5 GCM ensemble**

Even though the CORDEX RCM ensemble consists of several simulations using different GCM-RCM combinations, it may not represent the full potential spread of the climate change signal. To investigate how well the CORDEX RCMs capture the variability within the greater CMIP5 GCM ensemble, the average changes in temperature and precipitation over the Fennoscandian domain (region A in Fig 1) are calculated. Figure 5 shows that the ensemble spread in the CMIP5 ensembles is larger than in the CORDEX RCM ensemble. Especially, the difference between the minimum and maximum is larger in the CMIP5 GCMs than in the CORDEX RCMs. This could not entirely be explained by differences in ensemble sizes. See for example the numbers for RCP8.5 in Fig. 5c, where the CMIP5 GCMs and the CORDEX RCMs show large differences in spread although the ensembles are of comparable sizes. In the case of RCP8.5, the 67 members in the CORDEX RCM ensemble are only using 7 unique GCMs and 11 RCMs, which is much less than the 25 unique GCMs in the full CMIP5 ensemble. When looking at an ensemble just consisting of the 9 GCMs (including different realisations) used to force the RCMs the spread is much smaller.

The ensemble means, however, are quite similar. In general, the two ensembles agree on the large-scale differences, and the choice of emission scenario is of greater importance than the construction of the ensemble (Fig 5). The result is the same even when looking at smaller regions within the domain (e.g. regions B, C and D in Fig 1). A conclusion is that the Euro-CORDEX ensemble well captures the mean climate change signal, but that the spread is limited compared to the CMIP5 ensemble.

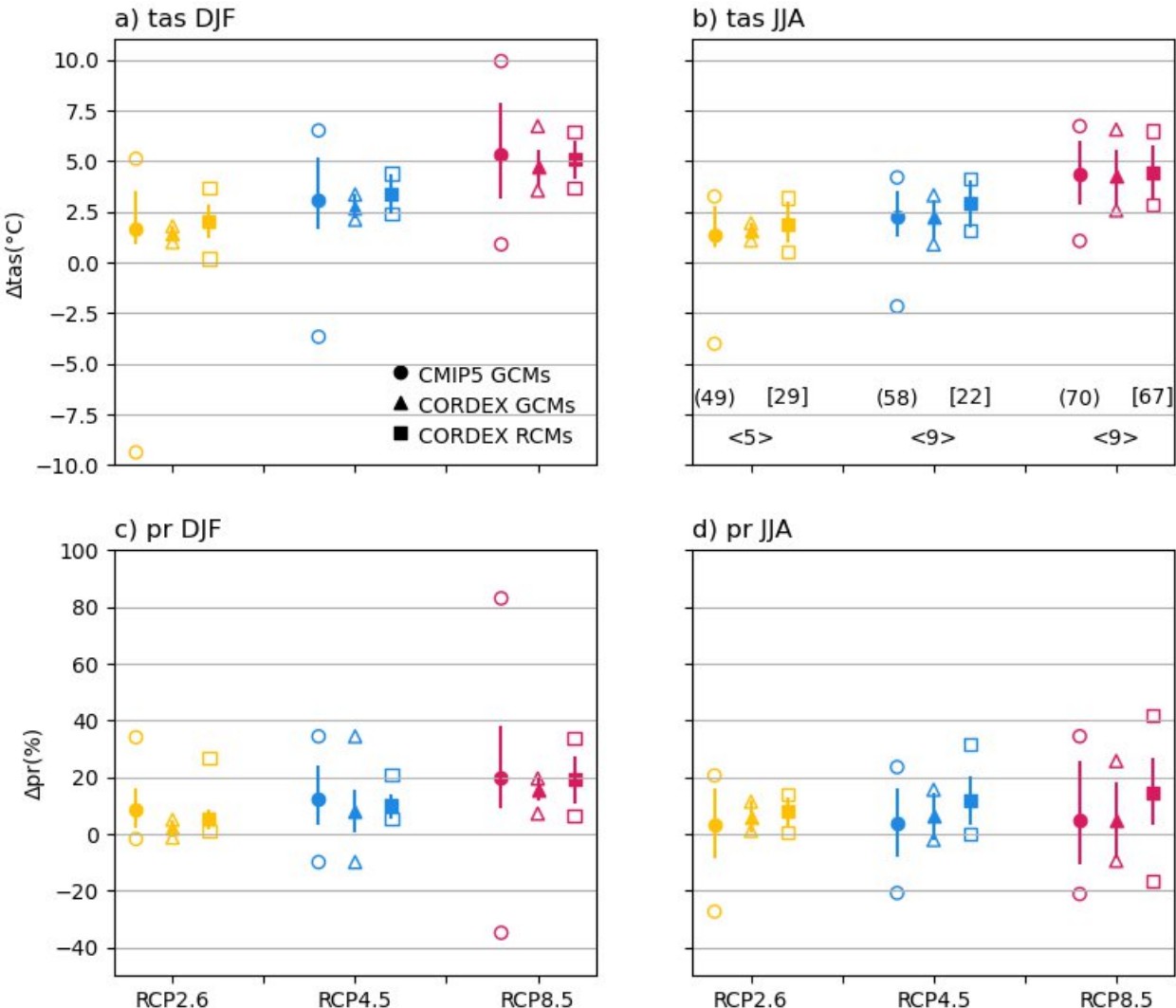

**Figure 5: Temperature (tas, °C) (a, b) and precipitation (pr, %) (c, d) anomalies in Fennoscandia 1971 — 2000 to 2071 — 2100 for winter (a, c) and summer (b, d) according to the scenarios RCP2.6 (green), RCP4.5 (yellow) and RCP8.5 (blue). The CMIP5 GCMs are represented by circles, the CORDEX GCMs by triangles and the CORDEX RCMs by squares. The central marker represents the ensemble mean, the line spans between the 10th and 90th percentiles, open markers show ensemble minima and maxima. Panel b) also shows the number of members in the respective ensembles.**

**3.5 How the simulated GWL climate is influenced by the choice of GCMs, RCMs and RCPs**

Here, we investigate how the characteristics of a certain GWL is influenced by the models and scenarios it is made of. Are all GWL2 the same, even if different models and scenarios are used to calculate them? First, we look at sub-ensembles based on GCMs (all members in a sub-ensemble are forced with the same GCM). Then we look at sub-ensembles based on RCM

and RCP (all members of a sub-ensemble used the same RCM and RCP, respectively). Statistically significant differences are assessed using an ANOVA analysis (see section 2.5).

### 3.5.1 Sub-ensembles based on driving GCMs

The results for sub-ensembles forced by the same GCM (all members of a sub-ensemble are forced with the same GCM, see Methods) are exemplified by temperature (tas) and annual number of summer days (su, see table 2 for definitions). Figure 6 shows which sub-ensembles that are significantly different from each other in the case of tas. All sub-ensembles from 1 to 7 are compared pairwise to see if they are significantly different or not. As an example, a green box at row 5 and column 1 means that sub-ensembles 5 and 1 are significantly different. In winter the average temperature change at GWL2 is +1.5-2.8 °C in the south and +1.7-4.2 °C in the north, depending on the chosen sub-ensemble (Fig. S9). Despite the rather large spread in warming the significant differences between sub-ensembles are not systematic in winter. However, in summer, where the temperature change is +1.0-2.5 °C in the south and 1.3-2.9 °C in the north (Fig. S9), there are systematic significant differences between sub-ensembles. The two sub-ensembles with the largest warming, labelled 4 & 7, are significantly different from the other sub-ensembles (green boxes at lines 4 and 7, and columns 4 and 7 in Fig 5). This pattern is also, to some extent, seen for su (Fig 7). In the south, sub-ensemble 7 is significantly different from 5 of the other sub-ensembles; in the north sub-ensemble 4 is significantly different from 5 other. For precipitation, the difference at GWL2 is small compared to the variability within each sub-ensemble. Only a few pairs of sub-ensembles are significantly different (none in summer in the north), but not in a systematic way (Fig. S10).

The choice of GCM can have a large impact on the ensemble. The difference in simulated change in tas can be up to 2 °C depending on the driving GCM; this does however, transfer into consistent significant differences for only two sub-ensembles.

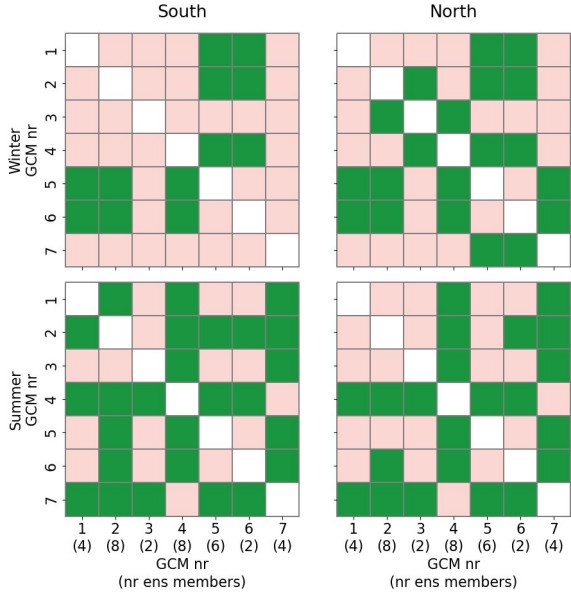

**Fig 6: Matrix of significant differences in temperature (tas) between GCM-based sub-ensembles within RCM17, for southern Sweden (South, region C in Fig. 1) and northern Sweden (North, region D in Fig. 1). Green colours indicate significant differences between two sub-ensembles and pink non-significant differences. White colours indicate that an ensemble is compared with itself. Numbers indicate sub-ensemble numbers, with the number of members in parenthesis.**

345

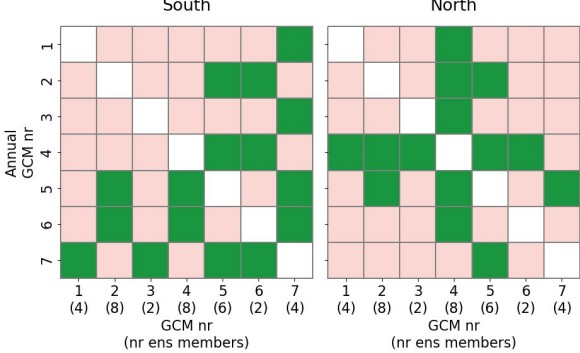

**Figure 7: Same as Figure 6 but for annual number of summer days (su, see table 2 for definitions)**

### 3.5.2 Sub-ensembles based on RCMs

Then, we proceed looking at sub-ensembles where the same RCM is used (all members of a sub-ensemble use the same RCM). Figure 8 shows which sub-ensembles that are significantly different from each other with regards to tas. The difference in change is about 1 °C between the sub-ensemble with the smallest and the largest change. Still, sub-ensemble no. 7 is the only sub-ensemble with systematically significant differences; in winter in the northern region and in summer it's different to all, or all but one, of the other ensembles. Sub-ensemble no. 7 is the sub-ensemble with the smallest temperature

increase. For su there are more significant differences in the southern region than in the northern, reflecting the larger variability in su in the south (Fig 9). There are however, only two sub-ensembles that are significantly different to three other sub-ensembles. Again, sub-ensembles 4 and 7, with a low number of su. For precipitation, the difference at GWL2 is small compared to the variability within each sub-ensemble. Only a few pairs of sub-ensembles are significantly different (in winter in the north one), but not in a systematic way (Fig. S11).

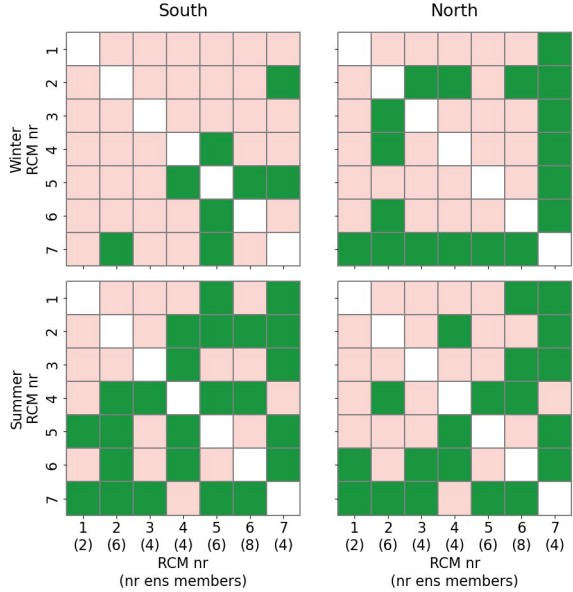

**Figure 8 Matrix of significant differences in temperature (tas) between RCM-based sub-ensembles within RCM17, for southern Sweden (South, region C in Fig. 1) and northern Sweden (North, region D in Fig. 1.). Green colours indicate significant differences. Numbers indicate sub-ensemble numbers, with the number of members in parenthesis.**

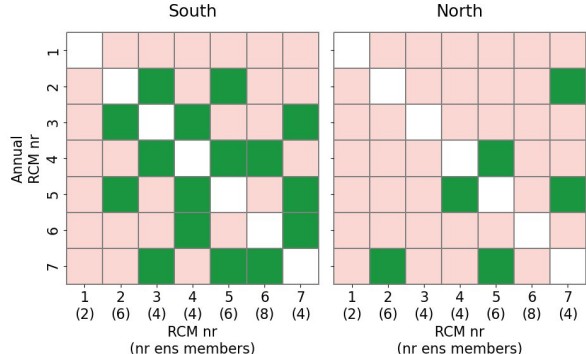

**Fig 9 Same as Fig 8, but for annual su.**

### 3.5.3 Sub-ensembles based on RCPs

As a last step, we look at sub-ensembles using the same RCPs. This analysis answers the question whether it matters which RCP you use to describe a GWL climate. In this case there are only two sub-ensembles to be compared. The differences between the ensembles based on RCP4.5 and RCP8.5 are generally small and not significant (see fig S12 for tas). RCP8.5 give larger anomalies in tas, tasmin and tasmax in summer in all regions. The difference compared to RCP4.5 is around 0.15°C and just below the 95 % confidence interval. The difference in all other indicators are insignificant on the 99 % level.

Inevitably, the characteristics of a climate model ensemble is determined by the simulations that it consists of. Using other models will not give the same results. These differences are however not systematic in any way, and mostly not significant. Even though an ensemble should be constructed with care, the role of the composition should not be exaggerated.

## 4 Discussion

### 4.1 The role of the models used on projected climate change

The projections of future climate presented here are consistent with other studies of the European climate (e.g. Coppola et al., 2021; Ranasinghe et al., 2021) and the climate in the Nordic region (e.g. Christensen et al., 2022). No pre-selection of models was made, which makes the ensemble used here an unbalanced 'ensemble of opportunity'. In such cases there is a risk that some models are under- or over-represented, which influences the ensemble mean (Evin et al., 2021; Sobolowski et al.,2025). On the other hand, information is lost when simulations are discarded, and natural variability is best sampled by single-model large ensembles (e.g. von Trentini et al., 2019; Maher et al., 2020). Furthermore, we note that different selections of individual GCM-RCM-RCP-combinations can have significant impact of the resulting ensemble as illustrated above. In the end, it is difficult to say that there is one approach that is always the most suitable. Different choices in the construction of an ensemble can be made and motivated depending on the aim.

Insufficient aerosol forcing is proposed as a reason for the observed underestimation of the trend in summer temperature in RCMs over central Europe compared to observations (e.g. Boé et al., 2020; Schumacher et al., 2024). However; the difference in summer warming between CORDEX and ERA5 is small in southern Sweden and Finland, and actually positive in Norway and northern Sweden (Schumacher et al., 2024). Bias adjustment may change the climate change signal. This is, however, generally seen as an improvement of the signal (Gobiet et al., 2015). MIdAS, the bias adjustment method used here, is shown to add a small increase in the climate change signal for both temperature and precipitation in Europe (Berg et al., 2022). The effect of bias adjustment on indicators is unknown and should be studied in the future.

A notable feature in the scaling between local and global climate change is seen for the precipitation indicators (Figs 4g & h). Here, there are clear differences between RCP4.5 and RCP8.5 even at the same level of global warming. It is previously

shown on the global scale that the response in precipitation depends on both surface warming and radiative effect of increased amounts of greenhouse gases (Pendergrass et al., 2015). The net effect of these depends on the RCP scenario. Furthermore, the aerosol forcing is different in the different scenarios. This would make GWLs less suitable for precipitation. On the European scale this is further complicated by local features. The weaker response in precipitation could be a consequence of drier conditions over the European continent leading to excessive evaporation and soil drying (e.g. Tuel and Eltahir, 2021).

## 4.2 Difference in model spread between GCM and RCM ensembles

In this study we show that the spread between the driving GCMs were larger than the spread between RCMs, even in the cases when the RCM ensemble had more members. This is supported by Kjellström et al. (2018). A potential explanation is that number of members is not the same as number of models. Previous studies show that multi-model ensembles have larger spread than single-model ensembles of similar, or even larger, sizes (von Trentini et al., 2019; Maher et al., 2021). This is perhaps not surprising. As different models have different physics a multi-model ensemble can offer a wider response to forcing and natural variability than what a single-model ensemble can. A support to this is that the ensemble means in the CORDEX GCM ensemble is not affected in any major way when we include more realisations with some GCMs. Likely, adding more realisations gives a better estimate of natural variability and extremes, but does not influence the mean values as much, since all realisations simulate the same climate (as opposed to simulations with different physics or forcing).

In this study bias adjusted RCMs are compared to non-adjusted GCMs. Bias adjustment may reduce the model spread in absolute values since systematic biases are minimised and all models are forced towards the reference data. The model spread in the climate change signal would, however, not be affected, assuming that bias adjustment with MIdAS preserves the climate change signal (Berg et al., 2022). The analysis of model spread in Section 3.5 and Figure 5 builds on the spread in climate change signal. Consequently, the difference between GCMs and RCMs are likely not explained by the application of bias adjustment.

Another explanation for differences in model spread is inconsistencies in forcing between the RCMs and the driving GCMs, where aerosol forcing probably is the most prominent factor in the context of this study (Taranu et al., 2023). This problem is indeed seen in both GCMs and RCMs, but only for summer in central Europe (Schumacher et al., 2024).

## 4.3 On the characteristics of GWL ensembles

As GWLs in fact are used for many different purposes it is necessary to investigate the characteristics of GWL ensembles. Especially how RCPs influence the GWL climate. Our study shows, for a broad range of indicators, that the choice of RCPs used has minimal effect on the GWL climate created. Furthermore, it is difficult to show that the inclusion of specific GCMs and RCMs influence the GWL climate in a significant way. This is perhaps expected considering that GCMs and RCMs are not independent (Sørland et al., 2018) and that the uncertainty in climate change due to GCMs can be as large as the uncertainty due to RCMs (Evin et al., 2021).

A caveat to our findings relates to the small number of members in the sub-ensembles. Sizes of 2-8 make it difficult to draw robust conclusions. Small samples reduce the power of the ANOVA test to detect differences between sub-ensembles and

more likely to fail to reject a false null hypothesis. In any case, this—and similar—ensemble is what is used to create GWL ensembles, and they must therefore be evaluated as much as possible. Adding more members would increase the statistical power, but would also change the ensemble to something else. We just have to do what we can with the ensemble at hand. A more solid evaluation could perhaps be achieved if AI or emulators were first used to fill all gaps in the matrix. That would enable a balanced comparison across GCMs and RCMs.

We performed our analysis on GWL1.5 and GWL2 and our conclusions only apply to these specific GWLs. It would be interesting to expand the analysis to more GWLs, but there are practical limitations to this. Smaller GWL increments would mean larger overlap between GWLs making it difficult to draw robust conclusions about the differences between GWLs. Furthermore, most RCP4.5 simulations do not reach GWL3 which means that the ensemble size would be heavily reduced, making the statistical analysis less solid. Also, if only one RCP reaches GWL3 it is not possible to investigate the role of

RCPs in the construction of a GWL; perhaps the most relevant thing to know. To study a range of GWLs in a RCM ensemble is another study, a study that would require other simulations, and maybe simulations that do not exist (for example more scenarios that reach GWL3).

## 5 Summary and conclusions

Global warming means for Fennoscandia higher temperature, more warm days and fewer cold days. In southern Sweden the number of summer days is doubled until the end of the century according to RCP4.5. At the same time the number of frost days decreases with 20-50 %. Precipitation increases generally, this shows in increasing mean precipitation, increasing number of days with heavy precipitation and decreasing number of dry days.

The RCM ensemble used here capture, on average, the change pattern from the CMIP5 GCM ensemble. The ensemble

spread, however, is larger in the CMIP5 ensemble.

The choice of RCP has minimal significance on the GWL2 ensembles. This implies that it would be safe to mix RCPs in the construction of GWL ensembles in order to increase ensemble size, and that a GWL could be based on only one RCP. It should be noted however, that we only look at mean changes. Trends within a GWL period do indeed depend on the RCP, and this could influence extremes. For example: the last years within a GWL period based on RCP8.5 may be warmer than

the last years within a GWL period based on RCP2.6. The largest difference between GWL2 sub-ensembles, regardless of how they are constructed in terms of combining GCMs and RCMs, is seen for temperature-based indices; however, it is difficult to say whether the choice of GCM or RCM contributes most to these variations.

All studied climate indicators scale somewhat linearly to the change in GMST. For indicators based on temperature thresholds, there may be a shift in trend slope when the temperatures rise above a certain level. Currently the regional

temperature change in Sweden is almost twice as large as the global trend. This ratio will decrease as GMST increases, to more and more approach a one-to-one relationship. This suggests that there is a limit to the feedback mechanisms that now accelerates the warming in Sweden. And that the ratio between local and global warming currently may be at its largest. Furthermore, this means that the steady relationship between global and regional warming that is sometimes assumed in weather attribution and regional warming levels may not hold in the future.

## Acknowledgements

We acknowledge Euro-CORDEX and all groups that contribute with simulations. Analyses were performed on the Swedish climate computing resources Bi and Freja at the Swedish National Supercomputing Centre (NSC), Linköping University. This is a contribution to the strategic research areas MERGE (ModElling the Regional and Global Earth system) and the Bolin Centre for Climate Research.

## Funding statement

This work is partially funded by Sweden's innovation agency, Vinnova, to the project titled "Adapting Urban Rail Infrastructure to Climate Change (AdaptUrbanRail)" (grant no. 2021-02456). This work was supported via Horizon Europe IA project Precilience (Grant Agreement 101157094).

## Author contributions

**GS** – Conceptualization, Methodology, Formal analysis, Visualization, Project administration, Writing – Original Draft; **AT** – Methodology, Formal analysis, Visualization, Writing – Review & Editing; **LB** – Software, Data Curation, Writing – Review & Editing; **EK** – Writing – Review & Editing; **MS** - Data Curation, Writing – Review & Editing; **RW** – Software, Data Curation, Writing – Review & Editing; **GN** – Resources

## Data availability

Ensemble means of 30 year periods (absolute values and anomalies) can be viewed and downloaded at: https://www.smhi.se/en/climate/tools-and-inspiration/climate-change-scenario/climate-change-scenario-tool

## Competing interests

The authors declare that they have no conflict of interest.

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
