# Peer review of "Projected climate change in Fennoscandia — and its relation to ensemble spread and global trends"

_EGUsphere, 2025_

## Referee Comment (RC2)

The manuscript by Strandberg et al. analyses an ensemble of CORDEX simulations for a variety of climate indicator over Fennoscandia. The influence of the GCM, RCM and RCP selection on future projections is analyzed. Further, the similarity of different sub-ensembles based on the same GCM/RCM/RCP at a global warming level of +2°C is assessed. Lastly, the slope of trends is assessed with respect to the trend in global mean temperature.

Analyzing the validity of using GWLs for various metrics and the influence of GCM/RCM/RCP choices on the projected trends is of relevance.

**Major concerns**

1) One of your key analyses is the influence of the ensemble sub-setting on mean trends in various metrics. However, the base ensemble is too small to create sufficiently large sub-ensembles to draw robust conclusions from the comparison. For example, some of your sub-ensembles only consist of two realizations.

2) In the same way, your conclusion on the possibility to mix emission scenarios when using global warming levels, is only based on the analysis of a single global warming level (GWL2). You would have the possibility to analyze a larger range of GWLs and test whether this holds true across all GWLs or whether at some point the mixing emission scenarios becomes a problem. When you choose smaller GWL increments, then you can also include the rcp2.6 simulations.

3) In my opinion a clear discussion is missing. The discussion has partly been implemented into the results; however, I think the following key points require a dedicated discussion:

    a. how the pre-selection has influenced your sub-ensembles, meaning there have been reasons for not using specific GCMs for downscaling, i.e. for example generally bad performance of the GCM.

    b. the new insights on the aerosol problem in RCMs (see Schumacher et al 2024 (https://doi.org/10.1038/s43247-024-01332-8)); this will influence your results on local vs. global trends

    c. the influence of the bias-correction on your trends. I am not familiar with the bias adjustment scheme, so a discussion on whether the method is trend preserving or not is required. In this regard you might also want to check the raw RCM ensemble in comparison to the GCM ensemble. The bias adjustment likely reduces the spread of your ensemble.

    d. discussion on the validity to use GWLs for precipitation (e.g., maybe see Pendergrass et al. 2015 (https://doi.org/10.1002/2015GL065854) and Gampe et al 2024 (https://doi.org/10.5194/esd-15-589-2024))

    e. discuss the reasons for the large difference between the CMIP5 ensemble and the CORDEX ensemble

    f. how robust is your ANOVA analysis in the very small sub-ensembles?

    g. Generally, place your findings within the existing literature. Both, your projections and the influence of the GCM or RCM choice, as well as the use of GWLs instead of time. (e.g., Evin et al 2021 (https://doi.org/10.5194/esd-12-1543-2021), Sorland et al 2018 (https://iopscience.iop.org/article/10.1088/1748-9326/aacc77), Sobolowski et al 2025 (https://doi.org/10.1175/BAMS-D-23-0189.1), Christensen et al 2022 (https://doi.org/10.5194/esd-13-133-2022))

h. Discuss the influence of natural climate variability on your results. You discuss the uncertainty from GCM, RCM and RCP, however, the uncertainty of natural climate variability can on local scales be an important source of uncertainty.

**General comments:**

1) The scientific merit of your paper needs to be better presented. The paper is based on a previous iteration of analysis. While model generations and spatial resolution are mentioned as an advancement, the clear advancement over the older paper is the analysis of global warming levels and the influence of the ensemble selection. This should be mentioned more prominently.

2) Further, the introduction can profit from the following additions:
   a. At the end of the introduction extend your description of what the paper is about. Try to be more concrete. Connected to this, add a motivation for your research questions (why is it important to look at local vs. global trends? Why do we need to analyze the influence of ensemble selection?) and what the actual research gap is.
   b. a short paragraph on projected changes over Fennoscandia

3) The methods section needs to be extended to be more comprehensive and clearer.
   a. Your paper is based on both bias-adjusted RCMs and raw GCMs, however, it is not clear when and where you use which data. This could be better explained in the methods sections. Further, more information on the bias-adjustment itself is needed.
   b. Also, a better and more straightforward description of your ensemble subsetting is required. (see my suggestion further down)
   c. Several details are missing, e.g. reference period and justification why you used the old 1971-2000 period; how were the GWLs calculated

4) The figure captions and figure description in the text are often not very clear. It is often not clear what data we are looking at. The same for the figures in the supplementary.

5) The analysis in section 3.4 needs to be extended to more GWLs to back your statements.

6) You have calculated multiple indicators, however, there is no consistency throughout your analysis. You sometimes show all indicators, then focus the analysis on seasonal temperature/precipitation, in other cases on annual temperature and csu, then on all indicators again. I would suggest focusing on a few key indicators and perform all analysis on these indicators. This way we can consistently follow the influence of the different choices (e.g. rcm vs gcm, sub-ensembles, trends at different GWLs).

7) Find better section headings to clearly reflect the content of the section.

**Detailed comments:**

**Abstract**:

L 15: "The regional climate models capture the signal of the driving global models." This statement only tells half the story. Yes, they represent the trends of their respective driving GCMs, but are under representative of the model spread in CMIP5, which means that the RCM signals here are not representative of the possible model spread of available ensembles.

L16f: "This implies that it would be safe to mix emission scenarios [...]". I am not fully convinced that you can robustly draw this statement from your analysis. You have only analysed this for a single global warming level.

**Introduction:**
P2, L44: "model sensitivity to ..." Add model uncertainty to this list as well. For model uncertainty vs. natural variability see von Trentini et al. 2019 (https://doi.org/10.1007/s00382-019-04755-8).

P2, L47: "improvements include ..." The scientific merit of this study can be strengthened more. Now it sounds more like, we have added a few models and did bias correction compared to the previous iteration.

**Methods:**

**Table 1:**
Does this list only include the official EURO-Cordex runs or did you also include additional runs, such as from Reklies-De? This would fill the GCM-RCM matrix. Or the other way around why didn't you use the additional RCM runs to fill the GCM-RCM matrix and focus on even sub-ensemble sizes?
Reklies-De: https://swift.dkrz.de/v1/dkrz_26083c6525be4627aeecde1fffc2b977/ReKliEs-De/ReKliEs-De/Internet-ReKliEs-De/startseite.html#Regio; data available at the ESGF

**P5, Bias-adjustment:**
Can you please clarify whether only the RCM data was bias corrected or also the driving GCMs. Also when you compare the different ensembles CMIP5-full ensemble, CMIP5-Cordex-GCM, and Cordex-RCM mention which data you compare; bias adjusted RCM trends vs. GCM non-bias adjusted trends. For me this is not clear, but this might explain some of the differences that you find. I would suggest to also include the raw RCM data in your analysis.

P5, L77: "Midas ..." A bit more information on the bias adjustment method is required. For example, what is the training period; is the method trend preserving; does the day-of-year adjustment mean that for each day of the year adjustment factors were calculated.

P6, L82: "software package Climix" Include the footnote as a regular reference.

**Indicator selection:**
Since the indicator selection is partly based on stakeholder engagement it would be nice to learn more about the reasoning behind this selection. For example, could you give a few examples for which sectors or applications the indices are relevant for.

**Selection and analysis sub-ensembles:**
P7, L94-103: This entire section can be condensed. A table might convey this information more straightforward. A suggestion could be: Replace the table 1 with a matrix of GCM-RCM combinations, then either highlight for the three categories of sub-ensembles one example, e.g. highlight the row of a single GCM highlighting all RCMs that downscaled this GCM; or highlight a column with a single RCM highlighting that you are looking at the GCM spread; etc. If this is too cluttered, then have three matrices highlighting the sub-sampling method. This will make it much easier for the reader to grasp your sub setting strategy.

P7, L104: "We are here looking at GWL2, [...]" Why +2°C of warming? and why didn't you analyze multiple warming levels? Since you are discarding the rcp2.6 simulations from this analysis anyway, you could have looked at multiple warming levels. This would make your statements on "mean trends are the same at GWL2 despite the emission scenario" more robust and convincing. Also, I would here define your acronym GWL.

P7, L113: "[...] a region in northern Sweden and a region in southern Sweden (regions C and D in figure 1)" Why did you choose these two sub-regions for comparison?

P7, L113f: "The domain and time averaged data for each member together form an ensemble." This sentence is not well connected to the previous or the following sentence.

P7, L 114: "family-wise error" Please clarify what you mean with family-wise. This is the first time you use this terminology.

Fig1 caption: "full model domain" This is misleading. The Euro-CORDEX domain is larger than the black outline. Maybe also mention on which of these domains the bias-adjustment was performed.

**Results:**

**Figure 2:**
1) Please clarify which ensemble is used for the ensemble mean trends here.
2) Please also clarify whether this is based on bias-adjusted data or the raw projections
3) This is the first time you mention your reference period. Please include this information in the methods section. Further, why is the relatively old reference period chosen instead of the recent reference period (1991-2020)? Please include the reasoning for this choice of reference period; in the methods section.

P10, L 131f: "The annual warming [...]" Not clear what you mean by annual warming. Please clarify. Also, why don't you give the range of warming for rcp4.5?

P10, L 142: "more similar to" Language. maybe use "comparable to"

**Figure 3:**
1) Color scales are not very intuitive and are not colour-blind friendly. Maybe think about streamlining the colormaps. For each panel one needs to constantly check the colorbar to interpret the patterns. For example green can mean either a very small change (f) or very large changes (d,e,g).
2) Same comment as for Figure 1; Which ensemble? Bias-adjusted, yes/no.

**Section 3.3 (RCMs compared to GCMs and the larger CMIP ensemble)**
I would maybe recommend performing a bootstrapping of the large CMIP5 GCM, i.e. draw 5 (9) random GCM from the CMIP5 ensemble, then you can quantify the influence of the ensemble size on your results.

P12, L 177-181: "This could not entirely be [...] the spread is much smaller" This needs to be discussed in much more detail. For example, the large spread in winter in the GCMs can in parts be explained by the presence of large internal climate variability. See von Trentini et al. 2019 (https://doi.org/10.1007/s00382-019-04755-8) on the comparison of one RCM single model large ensemble compared to the full CORDEX multi-model ensemble. Also see Maher et al 2021 (https://doi.org/10.1038/s41467-020-20635-w) showing that the CMIP5 full ensemble captures

parts of the internal climate variability given by multiple single model large ensembles. Also please clarify which "67 members are only forced by 7 unique GCMs".

P12, L181-183: "Kjellström et al (2018) […]" Are your results also pointing to this conclusion? Or is this the reason for why the spread is smaller? If this is the latter, can you please elaborate more on this. Are there other studies that have a detailed look on the conservation of GCM to RCM signals? (e.g. Taranu et al 2022, https://doi.org/10.1007/s00382-022-06540-6)
Also, what needs to discussed is the influence of bias-adjustment on the model spread.

P12, L183: "Especially, the distance between the minimum and maximum larger" Language. Please rephrase, also larger than what?

P12, L184f: "the choice of emissions scenario is of greater importance than the construction of the ensemble" Where can we see this? Please include a figure reference.

**Figure 4:**
1) Why not simply show boxplots with whiskers extending to min-max?
2) It is not clear what we exactly see here. Are the triangles the non-bias adjusted driving GCMs or is it the GCM-RCM used in your small ensemble, what about the squares are these also GCMs or RCMs. Maybe it would be worth to also analyse the RCM runs here once based on the raw data and once on the bias adjusted data. This would allow for a cleaner separation of the influence of the GCM ensemble selection, the influence of the RCM, and whether the bias adjustment further reduces the spread.
3) I don't know how you constructed your CMIP5 ensemble, whether it contains multiple members per GCM or whether you only used a single member per GCM. In anyway, it might make sense to only compare a single realisation of each GCM here (ensemble of unique GCMs). The reason for this is, that contributions to the CMIP5 ensemble are not evenly spread meaning that some models, e.g. the CanESM2 has contributed multiple members and they might skew the spread to one or the other side.

**Section 3.4**

The section needs a better title that fits the content of the section. For example, "Agreement in trends at the same GWL"

P14, L 203: "it reduces the uncertainty around the choice of emission scenarios." Where can we see this? Otherwise please add some references, for example, these references might fit (e.g., Gampe et al 2024 (https://doi.org/10.5194/esd-15-589-2024))

P14, L206: "ensemble is sensitive to how it is constructed with regards to which models" This is however true for any ensemble also the one based on time.

P14, L210: "pair-wised compared" -> "compared pairwise "

**Section 3.5 and Figure 9:**

1) What is the exact purpose of this section? For panel Fig 9 panel a, I can see the purpose, we see a stronger local warming than for global temperature. This fits the section header. However, for all other indicators the relationship between the x-axis and y-axis is not straightforward. We can't compare whether the local trends are stronger/weaker than global trends. If your goal is to compare the local trends vs. global trends at the same GWL, then I would suggest calculating the global mean of the respective indicator and compare the global mean trend with the local mean trend. The different dots would

then be different GWL, e.g., 1, 1.5, 2, 2.5, etc. For frost days or nzero it might make sense to calculate global trends on the same latitudes as your Scandinavian domain.

2) The figure caption and the text in the manuscript lack some description of what we actually see. Are the dots ensemble mean change in RCMs vs. the change in GCMs? Which GCMs, the CMIP5 ensemble mean GMST or the driving GCMs of the RCMs?

3) Why did you move to the time period approach again? I thought your motivation is to use GWLs instead of time.

P18, L73: "Figs 8a-c" I think this should say Fig 9a-c, right?

**Summary and conclusions**

P18, L290: "Trends within a GWL period" What do you mean by this? Please clarify.

P18, L291-293: "The largest difference […] most to these variations" What about the bias correction method? Could this influence the trends as well? Did you also bias correct the GCMs or only the RCMs? especially for absolute temperature thresholds bias adjusted vs. non-bias adjusted makes a difference.

P19, L297-298: "This suggests that […]. And that the ratio […]" This needs to be discussed in more detail. Especially, in the light of the aerosol implementation in the RCMs.

---

## Author Comment (AC1)

Thanks to all three reviewers that made a great effort in commenting. This will certainly improve the manuscript. All comments and replies follow below (*replies in italics and red*).

**Comments from reviewer #1**

Review of 'Projected climate change in Fennoscandia — and its relation to ensemble spread and global trends' by Strandberg et al.

The manuscript presents results how temperature and precipitation change under projected climate change over Fennoscandia and Sweden using an ensemble CORDEX regional models with different GCMs, RCMs and three different rcp scenarios. The results provide important information about climate change in this region and should be published if the authors consider some major concerns described below. My main concern is that the Method section is not clearly presented, and several results are introduced without sufficient explanation of how they were calculated. For the findings to be properly understood and evaluated, methods should be linked to the results

*We thank the reviewer for a thorough review and helpful comments. Specific answers to the comments are given below. The manuscript has been restructured, a new discussion section is added and several sentences are rephrased to increase clarity. We have addressed all comments noting the reviewers concerns and adjusted the methods section. We hope that this in a sufficient way answers to the reviewers comments.*

The Introduction is generally well written. The choice of RCMs (CORDEX )is described, however there are some known issues with CORDEX data that should be mentioned, e.g. lack of aerosol impacts, precipitation and temperature biases (Vautard et al., 2021).

*We added text to mention this in section 2.1 where the CORDEX ensemble is described:*

*"This does not mean that the CORDEX simulations are without systematic errors. Vautard et al (2021) conclude that the simulations are generally too wet, too cold and too windy compared to observations. Some of the discrepancies between GCMs and RCMs, as well as the weak warming trend, could be explained by a too simple description of aerosol forcing (Boé et al., 2021; Katragkou et al., 2024)."*

The Method section should be revised to enhance clarity and allow for a better understanding of the results. What variables are bias corrected, and what are the results of this correction. Over what period is this correction performed? How does it affect future projections? The effect of the bias correction is missing in the representation of the results further in the manuscript. If some models are biased corrected more than others, I guess this would impact the comparisons. If it is described and represented in Berg et al (missing in references) the results should be briefly presented. Is it only over land? (Region A covers the ocean as well)

*The part in Methods about bias adjustment has been expanded on describing Berg et al. (2022) (now in the reference list) and with details on how the adjustment was made.*

*"Midas is based on quantile mapping 'day-of-year' adjustments (Themeßl et al 2011; Wilcke et al., 2013). This means that the distribution used to adjust the data is different for each day of the year. Midas is aiming at preserving the trend in future projections and does perform similar to methods that explicitly preserve trends (Berg et al., 2022)."*

*"The bias adjustment was made using the period 1980-2000 as a reference. The variables tas, tasmin, tasmax and pr (see Table 2 for explanations) were adjusted in all gridpoints within the domain. Note that any further mentions of the CORDEX RCMs refers to this bias adjusted ensemble covering all land and ocean grid points in Fennoscandia and the Baltic states (region A in Fig 1)."*

The selection and analyses of sub-ensembles section is also unclear. Is it only these 17 models that is used further in the manuscript? What are these 17 models used for? This is the first mention of GWL (not written out). How is the GWL found? In the GCMs that drive the RCM? This should be described better. How and when GWL 2 is reached in different models and projections should be included in a Figure. Only results for tas and csu are presented in the manuscript, while several are stated here.

*The 17 models are used for the analysis of sub-ensembles in section 3.4 (now 3.3) and nothing more. For the rest of the paper the full ensembles are used. To explain this and to give an explanation of how the GWLs are calculated we re-wrote the start of section 2.4 (now 2.5)*

*"We want to investigate the robustness of the ensembles and how the simulated climate at a specific global warming level (GWL) is affected by the choice of emissions scenario, and global and regional models. The GWLs are calculated based on the GMST using the period 1850-1900 as a reference, following the protocol in the IPCC-WG1 Atlas (Iturbide et al., 2022). A GWL is reached when the GMST for a moving 20-year time window for the first time passes that level. For example: GWL2 occurs when GMST for the first time is 2°C more than in the reference period. The timing of a GWL is represented by a central year. In this study we use 30-year periods for each GWL stretching from 15 years before the central year to 14 years after."*

*We also made a more structured explanation of the different ensembles used in 2.2, 2.4 and 2.5*

*GWL is now written out the first time it is mentioned.*

*In the methods section we list all indicators that we analysed. As the results in 4.3 are mostly non-significant we choose to exemplify this with tas and csu only.*

The methods section should also include a brief description of which periods the results are calculated.

*We added this to 2.3:*

*"The indicators are presented as averages for the 30-year periods used in the SMHI web service (SMHI, 2025): the reference period 1971-2000 and the future periods 2011-2040, 2041-2070 and 2071-2100. WMO recommends 1961-1990 as reference period for descriptions of climate change (WMO, 2017), but since several RCM simulations start 1971, a compromise is to use 1971-2000."*

Figure 1, is region B used in any analysis, I did not find any further reference to this region.

*The "B" fell out but is now included again in:*

*"The result is the same even when looking at smaller regions within the domain (e.g. regions B, C and D in Fig 1)."*

Figure 2 and 3, Is this the mean over all the CORDEX models available or only the 17? The introduction and title of the manuscript focuses on the spread of ensembles that are not shown in this figure. Could this be included in the figure by shading be added where there is a large spread between the models?

*This part uses the full ensemble as stated in the beginning of section 3:*

*"Here, we start by describing average climate changes according to the CORDEX RCM ensemble. To understand these trends, they are then put in relation to the trend in GMST in the driving GCMs (CORDEX GCMs). This is followed by a comparison in ensemble spread between CORDEX RCMs, CORDEX GCMs and a larger ensemble of CMIP5 GCMs to see how much of the potential spread that may be lost by not using all available GCMs. Section 3 is concluded by an investigation of how the description of a GWL based on the RCM17 ensemble is influenced by the GCMs, RCMs and RCPs of which it is constructed."*

For climate adaptations, the change in seasonality of precipitation is important, as summers become drier and autumn wetter for crop security. Although mentioned, the results could be included in supplementary.

*We agree to that and include corresponding figures for winter, spring, summer and autumn in the supplementary.*

'The signal is not robust', this needs to be shown, maybe by showing the spread between the RCMs in the Figures.

*We think it is clear how we define robust since we write: "half of the ensemble members give increasing number of dry days, and half of the members decreasing."*

As a general comment, since supplementary material is already included, the results described in the manuscript should also be provided. Referring to results and then stating 'not shown' is inconsistent and reduces clarity for the reader

*Thanks for pointing this out. We have now added new figures to the supplementary and replaced all 'not shown' with references to these figures.*

The calculations in section 3.3 are not described in the method section. Are these also bias corrected? What models are included? Should also include CMIP6 models as they have been available for several years(CMIP7 results are available soon).

*We added text to explain how the GCM ensembles are calculated:*

*"The bias adjusted CORDEX RMCs are compared to two GCM ensembles.*

*CORDEX GCMs: consisting of the GCMs actually used to drive the RCMs (leftmost column in Table 1) (ensemble sizes 5, 9 and 9 for scenarios RCP2.6, RCP4.5 and RCP8.5 respectively). This ensemble includes several realisations for some GCMs since they are used to force RCMs.*

*CMIP5 GCMs: consisting of all CMIP5 models available on the Earth System Grid Federation, but restricted to one realisation per GCM to avoid overweight on certain GCMs (ensemble sizes 24, 28 and 34 for scenarios RCP2.6, RCP4.5 and RCP8.5 respectively)*

*The GCMs are not bias adjusted. For all GCMs the grid points falling within the Fennoscandian region (A in Figure 1) are used to calculate ensemble mean and spread for the region. For both GCM ensembles the global mean surface temperature (GMST) as 30-year averages for the reference period 1971-2000 and the future periods 2011-2040, 2041-2070 and 2071-2100 are calculated."*

*The point of this section is to investigate the relationship between the RCMs and the driving GCMs, as well as estimating how the model spread could change if more GCMs were used — the 'missing spread' if you will. With this perspective the CMIP6 models have no relevance for the comparison. Since they are not used to drive the RCMs here there is no point of including them.*

Figure 4 should have a legend, and darker color for the RCP4.5 and RCP8.5 models.

*A legend is added to panel a) in the figure, and the colours are changed.*

Figure 5- 8. These results would be more significant for future work if the different RCMs and GCMs were identified by model name on the y-axis. As some GCMs are too warm or too dry (or cold and wet) and labeling them would help highlight which models show large discrepancies.

*Remember that figures 5-8 only show whether the sub-ensembles are significantly different from each other, not whether certain models are too cold or too warm. Remember also that the results are combinations of GCMs and RCMs, it would therefore be difficult to evaluate the GCMs or RCMs based on these figure, even if the names where spelled out.*

Figure 9 Change colors (same as Figure 4) and the different time periods could have different markers.

*Changed as suggested.*

The calculations in section 3.5 is also not mentioned in Methods, how is the global temperature calculated?

*We have made it clearer by adding a new section 2.4*

*"The bias adjusted CORDEX RMCs are compared to two GCM ensembles.*

*CORDEX GCMs: consisting of the GCMs actually used to drive the RCMs (leftmost column in Table 1) (ensemble sizes 5, 9 and 9 for scenarios RCP2.6, RCP4.5 and RCP8.5 respectively). This ensemble includes several realisations for some GCMs since they are used to force RCMs.*

*CMIP5 GCMs: consisting of all CMIP5 models available on the Earth System Grid Federation, but restricted to one realisation per GCM to avoid overweight on certain GCMs (ensemble sizes 24, 28 and 34 for scenarios RCP2.6, RCP4.5 and RCP8.5 respectively)*

*The GCMs are not bias adjusted. For all GCMs the grid points falling within the Fennoscandian region (A in Figure 1) are used to calculate ensemble mean and spread for the region. For both GCM ensembles the global mean surface temperature (GMST) as 30-year averages for the reference period 1971-2000 and the future periods 2011-2040, 2041-2070 and 2071-2100 are calculated."*

*In 3.5 (now 3.3) we also make it more clear that CORDEX is compared to CMIP5: "In this section, we take a look at how local climate change in the CORDEX RCMs relates to the change in global mean surface temperature (GMST) in the CMIP5 GCMs (fig 4)."*

Need to justify the statement 'and that a GWL could be based on only one RCP'. This is not shown in the manuscript.

*This is supported by the previous sentence: "The choice of RCP has minimal significance on the GWL2 ensembles." Since the choice of RCP does not matter, a GWL could be constructed by one or many RCP and still be the same (with the caveat that we are only looking at mean values and not extremes).*

The references section should be checked. I could not find the Berg et al. 2022 reference and IPPC chapters should be referenced as they state on the first(second) page.

*Berg et al., 2022 is added, we do not see any problems with the references to IPCC.*

**Minor Comments**

P2. L38: Give some examples of the physical processes that is better represented.

*We added "local events like convective rain and short-duration extreme events" to the sentence.*

P2. L42-43: Give a reference for this statement.

*The following references were added*

*Déqué, M., Somot, S., Sanchez-Gomez, E. et al. The spread amongst ENSEMBLES regional scenarios: regional climate models, driving general circulation models and interannual variability. Clim Dyn 38, 951–964 (2012). https://doi.org/10.1007/s00382-011-1053-x*

*Coppola, E., et al.: Assessment of the European climate projections as simulated by the large EURO-CORDEX regional climate model ensemble J. Geophys. Res.: Atmospheres 126 e2019JD032356, 2021.*

p.2 l.49-51. This sentence could be improved

*This was rephrased to: "Climate model projections is an important tool for illustrating various aspects of climate change and how it could impact society. This data is used to support decision makers' work on climate change adaptation in Sweden."*

p3. L.67: CMIP5 has been written out previously.

*Corrected*

P5. L78-79: SMHIGridClim should not be in the same parenthesis as the reference (split), it is confusing if it is a reference or an abbreviation.

*Changed as suggested*

P6. Table 2 Could be simplified by putting 'season or year' in the legend of the table.

*The indicators are calculated as both annual and seasonal values. The text "year or season" seems redundant since it is implied that averages are made, and number of day are counted, for a selected period. Thus, this is removed. The only exception is tasmin and tasmax, where it is important to mention that e.g. tasmax is not the maximum of the daily maxima, but the average of daily maxima over a period. We change these texts to "The daily minimum/maximum temperatures averaged over a selected period". We hope that this also would simplify things.*

P7. L108. Split the parenthesis.

*Changed as suggested*

P11 Figure 3. Are these yearly means?

*Yes, this is now stated in the caption*

P12 l 183. Use another word than distance.

*Changed to difference*

P15. L 233: Use different word than instead. Both results are presented.

*Changed to: "Then, we proceed looking at..."*

**Comments from reviewer #2**

The manuscript by Strandberg et al. analyses an ensemble of CORDEX simulations for a variety of climate indicator over Fennoscandia. The influence of the GCM, RCM and RCP selection on future projections is analyzed. Further, the similarity of different sub-ensembles based on the same GCM/RCM/RCP at a global warming level of +2°C is assessed. Lastly, the slope of trends is assessed with respect to the trend in global mean temperature.

Analyzing the validity of using GWLs for various metrics and the influence of GCM/RCM/RCP choices on the projected trends is of relevance.

*We thank the reviewer for a thorough review and very constructive comments, that has helped to significantly improve the manuscript. The manuscript has been restructured, a new discussion section is added and several sentences are rephrased to increase clarity. Specific answers to the comments are given below.*

**Major concerns**

1) One of your key analyses is the influence of the ensemble sub-setting on mean trends in various metrics. However, the base ensemble is too small to create sufficiently large sub-ensembles to draw robust conclusions from the comparison. For example, some of your sub-ensembles only consist of two realizations.

*We elaborate in the new Discussion. A problem that pertains, however, is that the ensemble studied is an existing ensemble. Therefore, we cannot add members because that would mean that we investigate another ensemble than the once used in climate services, for example. What we can do, though, is to be more careful in our conclusions:*

> *"A caveat to our findings relates to the small number of members in the sub-ensembles. Sizes of 2-8 make it difficult to draw robust conclusions. Small samples reduce the power of the ANOVA test to detect differences between sub-ensembles and more likely to fail to reject a false null hypothesis. In any case, this—and similar—ensemble is what is used to create GWL ensembles, and they must therefore be evaluated as much as possible. Adding more members would increase the statistical power, but would also change the ensemble to something else. We just have to do what we can with the ensemble at hand. A more solid evaluation could perhaps be achieved if AI or emulators were first used to fill all gaps in the matrix. That would enable a balanced comparison across GCMs and RCMs."*

2) In the same way, your conclusion on the possibility to mix emission scenarios when using global warming levels, is only based on the analysis of a single global warming level (GWL2). You would have the possibility to analyze a larger range of GWLs and test whether this holds true across all GWLs or whether at some point the mixing emission scenarios becomes a problem. When you choose smaller GWL increments, then you can also include the rcp2.6 simulations.

*We elaborate also on this in the new Discussion:*

> *"We performed our analysis on GWL1.5 and GWL2 and our conclusions only apply to these specific GWLs. It would be interesting to expand the analysis to more GWLs, but there are practical limitations to this. Smaller GWL increments would mean larger overlap between GWLs making it difficult to draw robust conclusions about the differences between GWLs. Furthermore, most RCP4.5 simulations do not reach GWL3 which means that the ensemble size would be heavily reduced, making the statistical analysis less solid. Also, if only one*

3) In my opinion a clear discussion is missing. The discussion has partly been implemented into the results; however, I think the following key points require a dedicated discussion:

a. how the pre-selection has influenced your sub-ensembles, meaning there have been reasons for not using specific GCMs for downscaling, i.e. for example generally bad performance of the GCM.

b. the new insights on the aerosol problem in RCMs (see Schumacher et al 2024 (https://doi.org/10.1038/s43247-024-01332-8)); this will influence your results on local vs. global trends

c. the influence of the bias-correction on your trends. I am not familiar with the bias adjustment scheme, so a discussion on whether the method is trend preserving or not is required. In this regard you might also want to check the raw RCM ensemble in comparison to the GCM ensemble. The bias adjustment likely reduces the spread of your ensemble.

d. discussion on the validity to use GWLs for precipitation (e.g., maybe see Pendergrass et al. 2015 (https://doi.org/10.1002/2015GL065854) and Gampe et al 2024 (https://doi.org/10.5194/esd-15-589-2024))

e. discuss the reasons for the large difference between the CMIP5 ensemble and the CORDEX ensemble

f. how robust is your ANOVA analysis in the very small sub-ensembles?

g. Generally, place your findings within the existing literature. Both, your projections and the influence of the GCM or RCM choice, as well as the use of GWLs instead of time. (e.g., Evin et al 2021 (https://doi.org/10.5194/esd-12-1543-2021), Sorland et al 2018 (https://iopscience.iop.org/article/10.1088/1748-9326/aacc77), Sobolowski et al 2025 (https://doi.org/10.1175/BAMS-D-23-0189.1), Christensen et al 2022 (https://doi.org/10.5194/esd-13-133-2022))

h. Discuss the influence of natural climate variability on your results. You discuss the uncertainty from GCM, RCM and RCP, however, the uncertainty of natural climate variability can on local scales be an important source of uncertainty.

*Thanks for these concrete suggestions. We have added a new Discussion section. It is too lengthy to be cited in total, instead we just summarise the sub-sections.*

*The role of the models used on projected climate change. Here we discuss how our ensemble relates to previous research, ensemble construction, forcing (including aerosols), natural variability and how the bias adjustment may influence the results.*

*Difference in model spread between GCM and RCM ensembles. Here we discuss reasons for difference in ensemble spread within the CORDEX RCMs, CORDEX GCMs and CMIP5 GCMs. This is also related to previous studies.*

*On the characteristics of GWL ensembles. Here we discuss how the GWLs are influenced by the ensemble together with examples from other studies. We also discuss the statistical power of the ANOVA test.*

**General comments:**

1) The scientific merit of your paper needs to be better presented. The paper is based on a previous iteration of analysis. While model generations and spatial resolution are mentioned as an advancement, the clear advancement over the older paper is the analysis of global warming levels and the influence of the ensemble selection. This should be mentioned more prominently.

*The ensemble studied here is presented for the first time. Kjellström et al. (2016) present a completely different ensemble. It is not correct that this paper only presents incremental improvements. To make this clear we changed a few words in the following sentence: "Here we present a **new** dynamically downscaled ensemble of climate projections for Sweden. Compared to the previous **ensemble** (Kjellström et al., 2016)"*

*The scientific merits lies both in the first presentation of this ensemble, which we think we make clear with the new sentence above. Another merit is the analysis of spread, trends and GWLs. We hope that this is made clear in the new ending of the Introduction (see next comment).*

2) Further, the introduction can profit from the following additions:

a. At the end of the introduction extend your description of what the paper is about. Try to be more concrete. Connected to this, add a motivation for your research questions (why is it important to look at local vs. global trends? Why do we need to analyze the influence of ensemble selection?) and what the actual research gap is.

*We wrote a new ending of the Introduction that hopefully make this more clear:*

> *"Since these data cover Fennoscandia and the Baltic States, they may also be applicable to surrounding countries. They are based on RCP (Representative Concentration Pathways) scenarios and CMIP5 (Coupled Model Intercomparison Project Phase 5; Taylor et al., 2022) global models. The Swedish climate service (SMHI, 2025) relies on these data, and at least until a CMIP6-based downscaled ensemble becomes available, they will continue to be used.*

> *This RCM ensemble is already existing and used. Therefore, it is important to also discuss how the ensemble is constructed and how that influence the characteristics of the ensemble. This study aims at four general topics:*

> *i)    Projected climate change in Fennoscandia. This paper serves as a general overview of projected climate change in Sweden based on the best available material, making this the currently best projection of climate change in the region and a basis for further research and decision-making.*

> *ii)    How local trends in climate relate to global warming. Fennoscandia is known to have a warming trend that greatly exceeds the global trend, but still with a relatively linear relationship (C3S, 2024). It is, however, unknown if this relationship will persist in the future.*

> *iii)    Model spread in the RCM ensemble compared to the spread of the larger CMIP5 ensemble. Since the RCM ensemble is forced by a sub-set of available GCMs the model spread is potentially reduced. This would mean that information is lost in the RCM ensemble.*

*iv)   Since it is likely that global warming will reach +2 °C within this century, and since the Paris Agreement (UNFCCC, 2015) speaks of a keeping the temperature rise to below 2 °C, it is natural that descriptions of projected climate change are formed around a 2 degree warmer world. The question is how such 'global warming levels' are influenced by the climate models and emission scenarios used to calculate them."*

b. a short paragraph on projected changes over Fennoscandia

*We feel that it perhaps would be to get ahead of things to write in the introduction what we will show in the results.*

3) The methods section needs to be extended to be more comprehensive and clearer.

a. Your paper is based on both bias-adjusted RCMs and raw GCMs, however, it is not clear when and where you use which data. This could be better explained in the methods sections. Further, more information on the bias-adjustment itself is needed.

*The text is expanded to make this clearer. We added text to explain how the GCM ensembles are calculated:*

>*"The bias adjusted CORDEX RMCs are compared to two GCM ensembles.*
>
>*CORDEX GCMs: consisting of the GCMs actually used to drive the RCMs (leftmost column in Table 1) (ensemble sizes 5, 9 and 9 for scenarios RCP2.6, RCP4.5 and RCP8.5 respectively). This ensemble includes several realisations for some GCMs since they are used to force RCMs.*
>
>*CMIP5 GCMs: consisting of all CMIP5 models available on the Earth System Grid Federation, but restricted to one realisation per GCM to avoid overweight on certain GCMs (ensemble sizes 24, 28 and 34 for scenarios RCP2.6, RCP4.5 and RCP8.5 respectively)*
>
>*The GCMs are not bias adjusted. For all GCMs the grid points falling within the Fennoscandian region (A in Figure 1) are used to calculate ensemble mean and spread for the region. For both GCM ensembles the global mean surface temperature (GMST) as 30-year averages for the reference period 1971-2000 and the future periods 2011-2040, 2041-2070 and 2071-2100 are calculated."*

*We also expanded the start of Results*

>*"Here, we start by describing average climate changes according to the CORDEX RCM ensemble. To understand these trends, they are then put in relation to the trend in GMST in the driving GCMs (CORDEX GCMs). This is followed by a comparison in ensemble spread between CORDEX RCMs, CORDEX GCMs and a larger ensemble of CMIP5 GCMs to see how much of the potential spread that may be lost by not using all available GCMs. Section 3 is concluded by an investigation of how the description of a GWL based on the RCM17 ensemble is influenced by the GCMs, RCMs and RCPs of which it is constructed."*

*And more about bias adjustment*

>*"Midas is based on quantile mapping 'day-of-year' adjustments (Themeßl et al 2011; Wilcke et al., 2013). This means that the distribution used to adjust the data is different for each day*

*of the year. Midas is aiming at preserving the trend in future projections and does perform similar to methods that explicitly preserve trends (Berg et al., 2022).*

*The bias adjustment was made using the period 1980-2000 as a reference. The variables tas, tasmin, tasmax and pr (see Table 2 for explanations) were adjusted in all gridpoints within the domain. Note that any further mentions of the CORDEX RCMs refers to this bias adjusted ensemble covering Fennoscandia and the Baltic states (region A in Fig 1)."*

b. Also, a better and more straightforward description of your ensemble subsetting is required. (see my suggestion further down)

*See comments below*

c. Several details are missing, e.g. reference period and justification why you used the old 1971-2000 period; how were the GWLs calculated

*See comments below*

4) The figure captions and figure description in the text are often not very clear. It is often not clear what data we are looking at. The same for the figures in the supplementary.

*This has been made more clear.*

5) The analysis in section 3.4 needs to be extended to more GWLs to back your statements.

*We actually did the studies also for GWL1.5 (this will be stated more clearly). We think it is safe to assume that our conclusions hold at leas up to GWL2. It is true that with higher warming this may change. On the other hand, the number of available scenarios decrease with higher GWLs, which means that there are no scenarios to mix anyway. Already at GWL3 most RCP4.5 scenarios are discarded.*

*We have to stress that this paper is an investigation of an existing material. Since this is our starting-point we cannot choose which members to include in our ensemble, because that choice is already made. In a similar way, GWL2 holds a unique position because a large number of descriptions about the future climate are based on GWL2, and of course because GWL2 is mentioned in the Paris Agreement. Thus, it is relevant for us to investigate how appropriate it is to use a RCM ensemble like this to say something about GWL2. To study a range of GWLs in a RCM ensemble is another study, a study that would require other simulations, and maybe simulations that do not exist (for example more scenarios that reach GWL3). In any case we added this to the Discussion:*

*"We performed our analysis on GWL1.5 and GWL2 and our conclusions only apply to these specific GWLs. It would be interesting to expand the analysis to more GWLs, but there are practical limitations to this. Most RCP4.5 simulations do not reach GWL3 which means that the ensemble size would be heavily reduced, making the statistical analysis less solid. Also, if only one RCP reaches GWL3 it is not possible to investigate the role of RCPs in the construction of a GWL; perhaps the most relevant thing to know. To study a range of GWLs in a RCM ensemble is another study, a study that would require other simulations, and maybe simulations that do not exist (for example more scenarios that reach GWL3)."*

6) You have calculated multiple indicators, however, there is no consistency throughout your analysis. You sometimes show all indicators, then focus the analysis on seasonal temperature/precipitation, in other cases on annual temperature and csu, then on all indicators again. I would suggest focusing on a few key indicators and perform all analysis on these indicators. This way we can consistently follow the influence of the different choices (e.g. rcm vs gcm, sub-ensembles, trends at different GWLs).

*We realise that the order of the subsection was maybe not the most logical. We therefore change Result and Discussion to follow this structure.*

1) *Description of climate change in the region (CORDEX RCMs)*
2) *Relation between local change and GMST (CORDEX RCMs, CORDEX GCMs)*
3) *Spread in the ensembles compared to each other and to CMIP5 (CORDEX RCMs, CORDEX GCMs, CMIP5 GCMs)*
4) *How GWLs are influenced by GCM, RCM, RCP (CORDEX RCMs). We put this last although it is only based on CORDEX RCMs, because it is a bit special add-on.*

*The paper starts of with a description of climate change in the region based on 9 indicators. For tas we show seasonal values to get a more detailed picture of the difference between seasons, and because the behaviour of some indicators if one understand these differences. The remaining 8 indicators are show as annual values, although seasonal variations are mentioned for pr. We think this is reasonable. The other indicators that are relevant to show data for JJA and DJF (tasmin, tasmax, nzero, pr, r10, dd) are show in the supplementary.*

*When the trends in these indicators are compared to GMST, 8 indicators are used. We only discarded dd to get a symmetrical figure and because dd was the least interesting.*

*The analysis of sub-ensembles were made for all indicators. The material is quite extensive and many figures look the same — and often just show insignificant results. Thus, for brevity we decided to use tas and csu to represent our findings.*

*The comparison of spread in the ensembles are made only for tas and pr because all other indicators are a function of these. No need to multiply the number of figures.*

*All in all, we think that there is consistency. Yes, we made some decision not to show all indicators at all times, but that is only to be concise. It would be a shame to reduce this to just an investigation of tas and pr. The indicators are in fact used, and therefore they need to be described.*

7) Find better section headings to clearly reflect the content of the section.

*All headings under Section 3 are changed.*

**Detailed comments:**

Abstract:

L 15: "The regional climate models capture the signal of the driving global models." This statement only tells half the story. Yes, they represent the trends of their respective driving

GCMs, but are under representative of the model spread in CMIP5, which means that the RCM signals here are not representative of the possible model spread of available ensembles.

*True. We added this to hopefully make this clear: "Yet, the model spread is smaller than in the CMIP5 ensemble, which means that the RCMs not fully represent the potential model spread."*

L16f: "This implies that it would be safe to mix emission scenarios [...]". I am not fully convinced that you can robustly draw this statement from your analysis. You have only analysed this for a single global warming level.

*We actually did the studies also for GWL1.5 (this will be stated more clearly). We think it is safe to assume that our conclusions hold at least up to GWL2. It is true that with higher warming this may change. On the other hand, the number of available scenarios decrease with higher GWLs, which means that there are no scenarios to mix anyway. Already at GWL3 most RCP4.5 simulations are discarded. Nevertheless, we add +2°C to the sentence: "This implies that it would be safe to mix emission scenarios in calculations of global warming levels, at least up to +2°C, and as long as mean values are concerned."*

Introduction:

P2, L44: "model sensitivity to ..." Add model uncertainty to this list as well. For model uncertainty vs. natural variability see von Trentini et al. 2019 (https://doi.org/10.1007/s00382-019-04755-8).

*We added model uncertainty and a reference to von Trentini to the sentence.*

P2, L47: "improvements include ..." The scientific merit of this study can be strengthened more. Now it sounds more like, we have added a few models and did bias correction compared to the previous iteration.

*It is true that the scientific merit could be strengthened more. There is however a difference between data material and studies. These sentences describe the model ensemble. Compared to the previous ensembles the improvements are exactly what is listed here. The scientific merit lies in what we do with the data.*

Methods:

Table 1:

Does this list only include the official EURO-Cordex runs or did you also include additional runs, such as from Reklies-De? This would fill the GCM-RCM matrix. Or the other way around why didn't you use the additional RCM runs to fill the GCM-RCM matrix and focus on even sub-ensemble sizes?

Reklies-De: https://swift.dkrz.de/v1/dkrz_26083c6525be4627aeecde1fffc2b977/ReKliEs-De/ReKliEs-De/Internet-ReKliEs-De/startseite.html#Regio; data available at the ESGF

*The Reklies RCM simulations are included with one exception, everything driven by CanESM. These were the simulations that were available when the ensemble presented here was constructed. The ESD downscalings are not included.*

P5, Bias-adjustment: Can you please clarify whether only the RCM data was bias corrected or also the driving GCMs.

*The methods section is now expanded to explain this better (see below).*

Also when you compare the different ensembles CMIP5-full ensemble, CMIP5-Cordex-GCM, and Cordex-RCM mention which data you compare; bias adjusted RCM trends vs. GCM non-bias adjusted trends. For me this is not clear, but this might explain some of the differences that you find. I would suggest to also include the raw RCM data in your analysis.

*Text is added at numerous places to make this more clear.*

P5, L77: "Midas ..." A bit more information on the bias adjustment method is required. For example, what is the training period; is the method trend preserving; does the day-of-year adjustment mean that for each day of the year adjustment factors were calculated.

*The section on bias adjustment is expanded:*

> *"To minimise systematic errors the Euro-CORDEX ensemble was bias adjusted using the method "Multi-scale Bias Adjustment" available in Midas (Berg et al., 2022). Midas is based on quantile mapping 'day-of-year' adjustments (Themeßl et al 2011; Wilcke et al., 2013). This means that the distribution used to adjust the data is different for each day of the year. Midas is aiming at preserving the trend in future projections and does perform similar to methods that explicitly preserve trends (Berg et al., 2022). As reference data the SMHI gridded climatology (SMHIGridClim) data set (Andersson et al., 2021) was used. SMHIGridClim covers Fennoscandia and the Baltic states (region A in Fig 1), which means that the bias adjusted ensemble covers a smaller domain centred over Sweden, instead of the entire European domain. The bias adjustment was made using the period 1980-2000 as a reference. The variables tas, tasmin, tasmax and pr (see Table 2 for explanations) were adjusted in all gridpoints within the domain. Note that any further mentions of the CORDEX RCMs refers to this bias adjusted ensemble covering Fennoscandia and the Baltic states (region A in Fig 1)."*

P6, L82: "software package Climix" Include the footnote as a regular reference.

*Changed as suggested*

Indicator selection:

Since the indicator selection is partly based on stakeholder engagement it would be nice to learn more about the reasoning behind this selection. For example, could you give a few examples for which sectors or applications the indices are relevant for.

*We added this sentence: "The indicators are meant to be relevant for large parts of society, but agriculture (Strandberg et al., 2024a) and the energy sector (Strandberg et al., 2024b) have also been specifically targeted."*

Selection and analysis sub-ensembles:

P7, L94-103: This entire section can be condensed. A table might convey this information more straightforward. A suggestion could be: Replace the table 1 with a matrix of GCM-RCM combinations, then either highlight for the three categories of sub-ensembles one example, e.g. highlight the row of a single GCM highlighting all RCMs that downscaled this GCM; or highlight a column with a single RCM highlighting that you are looking at the GCM spread; etc. If this is too cluttered, then have three matrices highlighting the sub-sampling method. This will make it much easier for the reader to grasp your sub setting strategy.

*We struggled indeed quite a bit to find the best way to illustrate this. Apparently we did not reach all the way. We think that table 1 would be to cluttered if it were to include also this. We also thinks that Table 1 serves its purpose in the way it already looks. Instead, we created a new table and text to replace the old explanation. We hope this works better:*

*"To illustrate the procedure, we make the hypothetical case of three GCMs (GCM1-3) and three RCMs (RCM1-3) combined in different ways (Table 3). A sub-ensemble only using GCM1 would include all RCMs forced by GCM1, i.e. the simulations in row R1 in Table 3, i.e.three simulations. In the same way the sub-ensemble based on GCM2 consist of two simulations. Sub-ensembles using only one RCM use all simulations with one RCM forced by different GCMs, i.e. one of the columns C1-3. The sub-ensemble based on RCM1 has three simulations. Sub-ensembles based on one scenario use all simulations run with that scenario.*

*Table 3* Table 3 Hypothetical sketch of how three GCMs (GCM1-3) could be downscaled by three RCMs (RCM1-3) and how the sub-ensemble strategy works

|      |      | C1 RCM1 | C2 RCM2 | C3 RCM3 |
|------|------|---------|---------|---------|
| R1   | GCM1 | X       | X       | X       |
| R2   | GCM2 | X       |         | X       |
| R3   | GCM3 | X       |         |         |

P7, L104: "We are here looking at GWL2, [...]" Why +2°C of warming? and why didn't you analyze multiple warming levels? Since you are discarding the rcp2.6 simulations from this analysis anyway, you could have looked at multiple warming levels. This would make your statements on "mean trends are the same at GWL2 despite the emission scenario" more robust and convincing. Also, I would here define your acronym GWL.

*We actually did also look at GWL1.5. The text is now changed to: "We analysed GWL1.5 and GWL2. GWL1.5 is reached in all scenarios, while GWL2 is reached in RCP4.5 and RCP8.5, but not in RCP2.6. Already at GWL3 most of the RCP4.5 are discarded since they do neat reach*

*that level of warming. The lack of different scenarios and the smaller ensemble size makes GWL3 a less interesting case."*

*GWL is now spelled out at the first time it occurs.*

P7, L113: "[...] a region in northern Sweden and a region in southern Sweden (regions C and D in figure 1)" Why did you choose these two sub-regions for comparison?

*These regions are meant to represent the different climates of northern and southern Sweden. This is added to the sentence to be clear: "The tests are done for a region in northern Sweden and a region in southern Sweden representing the different climatic conditions in Sweden (regions C and D in figure 1)." We did try several regions, but since they were not all that different we decided that two were enough, and that these two work well to show the difference between the north and the south.*

P7, L113f: "The domain and time averaged data for each member together form an ensemble." This sentence is not well connected to the previous or the following sentence.

*We deemed this sentence redundant and decided to remove it.*

P7, L 114: "family-wise error" Please clarify what you mean with family-wise. This is the first time you use this terminology.

*This means that the probability of one or more false positives among all grid points cells is 5 % instead of a 5 % false positive rate in each individual grid point, if no correction is applied. This is now added to the text.*

Fig1 caption: "full model domain" This is misleading. The Euro-CORDEX domain is larger than the black outline. Maybe also mention on which of these domains the bias-adjustment was performed.

*We changed this to: "A) Fennoscandian region (black, full line) is the domain on which bias adjustment is applied"*

Results:

Figure 2:

1) Please clarify which ensemble is used for the ensemble mean trends here.

*"CORDEX RCMs" are now added to the caption.*

2) Please also clarify whether this is based on bias-adjusted data or the raw projections

*To set the context Section 3 now starts:*

*"Here, we start by describing average climate changes according to the CORDEX RCM ensemble. To understand these trends, they are then put in relation to the trend in GMST in the driving GCMs (CORDEX GCMs). This is followed by a comparison in ensemble spread between CORDEX RCMs, CORDEX GCMs and a larger ensemble of CMIP5 GCMs to see how much of the potential spread that may be lost by not using all available GCMs. Section 3 is concluded by an investigation of how the description of a GWL based on the RCM17 ensemble is influenced by the GCMs, RCMs and RCPs of which it is constructed."*

3) This is the first time you mention your reference period. Please include this information in the methods section. Further, why is the relatively old reference period chosen instead of the recent reference period (1991-2020)? Please include the reasoning for this choice of reference period; in the methods section.

*We added text to the methods section to explain this:*

*"The indicators are presented as 30-year averages for the 30-year periods used in the SMHI web service: reference period 1971-2000 and the future periods 2011-2040, 2041-2070 and 2071-2100. WMO recommends 1961-1990 as reference period for climate change (WMO, 2017), but since several RCM simulations start 1971, a compromise is to use 1971-2000."*

P10, L 131f: "The annual warming [...]" Not clear what you mean by annual warming. Please clarify. Also, why don't you give the range of warming for rcp4.5?

*We changed this to "The increase in annual mean temperature in Sweden" and added the warming range for RCP4.5*

P10, L 142: "more similar to" Language. maybe use "comparable to"

*Changed as suggested*

Figure 3:

1) Color scales are not very intuitive and are not colour-blind friendly. Maybe think about streamlining the colormaps. For each panel one needs to constantly check the colorbar to interpret the patterns. For example green can mean either a very small change (f) or very large changes (d,e,g).

*Figure 3 is now updated to be more intuitive and colour-blind friendly.*

2) Same comment as for Figure 1; Which ensemble? Bias-adjusted, yes/no.

*Same ensemble as in Fig 2. See comment above.*

Section 3.3 (RCMs compared to GCMs and the larger CMIP ensemble)

I would maybe recommend performing a bootstrapping of the large CMIP5 GCM, i.e. draw 5 (9) random GCM from the CMIP5 ensemble, then you can quantify the influence of the ensemble size on your results.

*We don't really see the point with this. The CORDEX-GCM ensembles is already 5-9 members large. Randomly selecting 5-9 other members would indeed give other results, but we don't expect them to be all that different. The purpose of this is to i) relate the spread in the RCM ensemble to the spread of the GCMs driving the RCMs, ii) relating the spread in the small CORDEX-GCM ensemble to the spread in the larger CMIP5-GCM ensemble iii) and investigating the potential spread by comparing the RCM ensemble to the CMIP5-GCM ensemble. We think that we are able to do that with the current shape of Fig 4.*

P12, L 177-181: "This could not entirely be [...] the spread is much smaller" This needs to be discussed in much more detail. For example, the large spread in winter in the GCMs can in parts be explained by the presence of large internal climate variability. See von Trentini et al. 2019 (https://doi.org/10.1007/s00382-019-04755-8) on the comparison of one RCM single model large ensemble compared to the full CORDEX multi-model ensemble. Also see Maher et al 2021 (https://doi.org/10.1038/s41467-020-20635-w) showing that the CMIP5 full ensemble captures parts of the internal climate variability given by multiple single model large ensembles.

*A new Discussion section is added to discuss this:*

*"In this study we show that the spread between the driving GCMs were larger than the spread between RCMs, even in the cases when the RCM ensemble had more members. This is supported by Kjellström et al. (2018). A potential explanation is that number of members is not the same as number of models. Previous studies show that multi-model ensembles have larger spread than single-model ensembles of similar, or even larger, sizes (von Trentini et al., 2019; Maher et al., 2021). This is perhaps not surprising, as different models have different physics a multi-model ensemble can offer a wider response to forcing and natural variability that a single-model ensemble can. A support to this is that the ensemble means in the CORDEX GCM ensemble is not affected in any major way when we include more realisations with some GCMs. Likely, adding more realisations gives a better estimate of natural variability and extremes, but does not influence the mean values as much, since all realisations simulate the same climate (as opposed to simulations with different physics or forcing).*

*In this study bias adjusted RCMs are compared to non-adjusted GCMs. Bias adjustment may reduce the model spread in absolute values since systematic biases are minimised. The model spread in the climate change signal would, however, not be affected, assuming that bias adjustment with Midas preserves the climate change signal (Berg et al., 2022). The analysis of model spread in Section 3.5 and Figure 5 builds on the spread in climate change signal. Consequently, the difference between GCMs and RCMs are likely not explained by the application of bias adjustment.*

*Another explanation for differences in model spread is inconsistencies in forcing between the RCMs and the driving GCMs, where aerosol forcing probably is the most prominent factor in the context of this study (Taranu et al., 2023). Although this problem is seen in both GCMs and RCMs (Schumacher et al., 2024)."*

Also please clarify which "67 members are only forced by 7 unique GCMs".

*The 67 members in the CORDEX RCM ensemble are only using 7 unique GCMs. This is now better explained (see above).*

P12, L181-183: "Kjellström et al (2018) [...]" Are your results also pointing to this conclusion? Or is this the reason for why the spread is smaller? If this is the latter, can you please elaborate more on this. Are there other studies that have a detailed look on the conservation of GCM to RCM signals? (e.g. Taranu et al 2022, https://doi.org/10.1007/s00382-022-06540-6)

*Yes, we use Kjellström et al. (2018) to support our own findings. Se also comment above.*

Also, what needs to discussed is the influence of bias-adjustment on the model spread.

*We added this to the discussion:*

> *"Bias adjustment may reduce the model spread in absolute values since systematic biases are minimised and all models are drawn/forced towards the reference data/observations. The model spread in the climate change signal would, however, not be affected, assuming that bias adjustment with Midas preserves the climate change signal (Berg et al., 2022). The analysis of model spread in Section 3.5 and Figure 5 builds on the spread in climate change signal. Consequently, the difference between GCMs and RCMs are likely not explained by the application of bias adjustment."*

P12, L183: "Especially, the distance between the minimum and maximum larger" Language. Please rephrase, also larger than what?

*Rephrased to: "Especially, the difference between the minimum and maximum is larger in the CMIP5 GCMs than in the CORDEX RMCs."*

P12, L184f: "the choice of emissions scenario is of greater importance than the construction of the ensemble" Where can we see this? Please include a figure reference.

*We added a reference to Figure 5.*

Figure 4:

1) Why not simply show boxplots with whiskers extending to min-max?

*The reason is that in this way it is easier to use markers that can help distinguishing between the different types of ensembles. For example, it is easier to make a legend. It is more colour-blind friendly to use both colours and markers.*

2) It is not clear what we exactly see here. Are the triangles the non-bias adjusted driving GCMs or is it the GCM-RCM used in your small ensemble, what about the squares are these also GCMs or RCMs. Maybe it would be worth to also analyse the RCM runs here once based on the raw data and once on the bias adjusted data. This would allow for a cleaner separation of

the influence of the GCM ensemble selection, the influence of the RCM, and whether the bias adjustment further reduces the spread.

*Circles: CMIP5 GCMs, triangles: CORDEX GCMs, squares: bias adjusted CORDEX RCMs. We added a legend to Fig 4 to make this more obvious.*

3) I don't know how you constructed your CMIP5 ensemble, whether it contains multiple members per GCM or whether you only used a single member per GCM. In anyway, it might make sense to only compare a single realisation of each GCM here (ensemble of unique GCMs). The reason for this is, that contributions to the CMIP5 ensemble are not evenly spread meaning that some models, e.g. the CanESM2 has contributed multiple members and they might skew the spread to one or the other side.

*The ensemble used includes multiple realisations for some GCMs. We did actually mak the same plot using only one realisation per GCM. This approach gives no decisive differences. Nevertheless, we see the point of showing only individual GCMs, not complicating things with several realisations. We have therefore updated the figure. It seems like large spread is achieved by many models, not by many realisations. We elaborate on this in the text. Using several realisation—and thereby inflating the ensemble—may give the false impression that a very large ensemble is needed to get large spread, although this does not seem to be the case.*

Section 3.4

The section needs a better title that fits the content of the section. For example, "Agreement in trends at the same GWL"

*This is now changed to: 3.5 How GWL climate is influenced by the choice of GCMs, RCMs and RCPs*

P14, L 203: "it reduces the uncertainty around the choice of emission scenarios." Where can we see this? Otherwise please add some references, for example, these references might fit (e.g., Gampe et al 2024 (https://doi.org/10.5194/esd-15-589-2024))

*We see it when we shift the uncertainty in outcome at a specific point in time to a specific outcome within a time period. We added a reference to Maule et al., 2017*

*Maule, C. F., Mendlik, T. and Christensen, O. B.: IMPACT2C - Quantifying projected impacts under 2°C warming Climate Services, 7, 3, 11, 2405-8807, https://doi.org/10.1016/j.cliser.2016.07.002, 2017*

P14, L206: "ensemble is sensitive to how it is constructed with regards to which models" This is however true for any ensemble also the one based on time.

*Yes, the point here is that using GWLs is sometimes used as way to get around this. Sometimes it look as like people don't think that they have to bother about the choice of simulations as long as they use GWLs. We added an 'also' in the sentence to show that this is not unique for GWL ensembles.*

P14, L210: "pair-wised compared" -> "compared pairwise "

*Change as suggested.*

Section 3.5 and Figure 9:

1) What is the exact purpose of this section? For panel Fig 9 panel a, I can see the purpose, we see a stronger local warming than for global temperature. This fits the section header. However, for all other indicators the relationship between the x-axis and y-axis is not straightforward. We can't compare whether the local trends are stronger/weaker than global trends. If your goal is to compare the local trends vs. global trends at the same GWL, then I would suggest calculating the global mean of the respective indicator and compare the global mean trend with the local mean trend. The different dots would then be different GWL, e.g., 1, 1.5, 2, 2.5, etc. For frost days or nzero it might make sense to calculate global trends on the same latitudes as your Scandinavian domain.

*The purpose is to relate local climate change to global warming. We hear for example that warming in the Nordic region is twice the global warming. In panel a) we can see that this is more or less correct at present, but that this relationship may change in the future. This is obviously the most straightforward comparison. In panels b) and c) we see that the trends are not the same as for tas, and therefore it is relevant to keep the one-to-one and two-to-one lines.*

*For the rest of the indicators, panels d-h), what we want to show is how these indicators relate to global warming. Since change in GMST is the most common measure of this, we argue that it is interesting to see how any indicator scale compared to GMST. Interesting things to note could for example be that all indicators do not scale linearly and that the different RCPs do not follow the same paths.*

2) The figure caption and the text in the manuscript lack some description of what we actually see. Are the dots ensemble mean change in RCMs vs. the change in GCMs? Which GCMs, the CMIP5 ensemble mean GMST or the driving GCMs of the RCMs?

*We rephrased the caption a bit:*

*Fig 9: Climate change in in the Fennoscandian domain (region A in Fig 1) in the CORDEX RCMs (y-axes) relative to the difference in global annual temperature in the CMIP5 GCMs (x-axes), relative to the period 1971 — 2000. Different indicators are calculated based on RCM data: a) mean temperature (tas, °C), b) minimum temperature (tasmin, °C), c) maximum temperature (tasmax, °C), d) no. of frost days (fd, days), e) no. of summer days (su, days), f) no. of days with zero crossings (nzero, days) g) precipitation (pr, mm mon-1) h) no. of days with heavy precipitation (r10mm, days). Markers represent the periods 1971 — 2000 (cross), 2011 — 2040 (triangle), 2041 — 2070 (square), 2071 — 2100 (circle) for emissions scenarios RCP2.6 (green), RCP4.5 (orange) and RCP8.5 (light blue). In panels a-c the one-to-one relationship is shown with a dashed line, and the two-to-one with a dotted line.*

3) Why did you move to the time period approach again? I thought your motivation is to use GWLs instead of time.

*We are not advocating for one or the other, we are simply investigating the properties of the RCM ensemble. Since GWLs is something that is used we want to also investigate the properties of GWL ensembles. The structure of the paper is changed to make this more logical (see comment above).*

P18, L73: "Figs 8a-c" I think this should say Fig 9a-c, right?

*Right. This is corrected.*

Summary and conclusions

P18, L290: "Trends within a GWL period" What do you mean by this? Please clarify.

*We mean that the trend within a 30-year GWL period is different depending on the scenario used. The trend is stronger in RCP8.5 than in RCP2.6, and that the difference between the beginning and the end of the period is larger. This could, for example, mean that a GWL period could contain more extremes if it is based on a larger proportion of RCP8.5 simulations.*

*We added a sentence to explain this more: "For example: the last years within a GWL period based on RCP8.5 may be warmer than the last years within a GWL period based on RCP2.6."*

P18, L291-293: "The largest difference [...] most to these variations" What about the bias correction method? Could this influence the trends as well? Did you also bias correct the GCMs or only the RCMs? especially for absolute temperature thresholds bias adjusted vs. non-bias adjusted makes a difference.

*We added text to discuss this:*

*"Bias adjustment may change the climate change signal. This is, however, generally seen as an improvement of the signal (Gobiet et al., 2015). Midas, the bias adjustment method used here, is shown to add a small increase in the climate change signal for both temperature and precipitation in Europe (Berg et al., 2022). The effect of bias adjustment on indicators is unknown and should be studied in the future."*

*"In this study bias adjusted RCMs are compared to non-adjusted GCMs. Bias adjustment may reduce the model spread in absolute values since systematic biases are minimised. The model spread in the climate change signal would, however, not be affected, assuming that bias adjustment with Midas preserves the climate change signal (Berg et al., 2022). The analysis of model spread in Section 3.5 and Figure 5 builds on the spread in climate change signal. Consequently, the difference between GCMs and RCMs are likely not explained by the application of bias adjustment."*

P19, L297-298: "This suggests that [...]. And that the ratio [...]" This needs to be discussed in more detail. Especially, in the light of the aerosol implementation in the RCMs.

*We added this to Discussion:*

*"Insufficient aerosol forcing is proposed as a reason for the observed underestimation of the trend in summer temperature in models compared to observations (e.g. Boé et al., 2020; Schumacher et al., 2024). However; the difference in summer warming between CORDEX and ERA5 is small in southern Sweden and Finland, and actually positive in Norway and northern Sweden (Schumacher et al., 2024). Thus, in our region of study the summer warming trend does not seem to be underestimated."*

**Comments from reviewer #3**

The paper "Projected climate change in Fennoscandia – and its relation to ensemble spread and global trends" presents temperature and precipitation projections for Scandinavia using bias-adjusted EURO-CORDEX GCM-RCM ensembles. It compares these with CMIP5 GCM signals and analyzes the influence of GCM, RCM, and emission scenarios on projection differences. The study finds that the EURO-CORDEX ensemble aligns well with the broader CMIP5 GCM ensemble and supports using global warming levels to reduce emission scenario uncertainty. It also notes that warming in Scandinavia since pre-industrial times has been about twice the global average, though this regional amplification is expected to decrease as global temperatures continue to rise.

*We thank the reviewer for a thorough review and helpful comments. Specific answers to the comments are given below. The manuscript has been restructured, a new discussion section is added and several sentences are rephrased to increase clarity. We hope that this in a sufficient way answers to the reviewers comments.*

**Major revisions**

*Bias-Adjusted RCM vs. Raw GCM Data:*

Comparing bias-adjusted RCM data with raw GCM data is methodologically inconsistent. To avoid misinterpretation due to the effects of bias adjustment, it would be more appropriate to compare raw RCM and raw GCM simulations directly. In this context, a more detailed description of the applied quantile-mapping approach would be valuable—for example, whether it is trend-preserving and whether it modifies climate change signals.

*The section about bias adjustment is expanded to better explain how the methods work.*

*The purpose of the RCM GCM comparison is the following. The data that we present is based on a limited number of GCMs downscaled by a limited number of RCMs. The question is how much of the potential spread that we lose when just using a selection of GCMs. What if we used more GCMs, how would that change our description of climate change? Since the RCMs are forced by the raw GCM data, and since the data presented, in for example climate services, is the adjusted RCM data, we think that this is the right comparison to do. This paper is about investigating an existing ensemble to see how it is influenced by the underlying choices, for example the choice of driving GCMs.*

*Sub-Ensemble Construction:*

The rationale behind the construction of the sub-ensembles remains unclear. The purpose and in particular the objectives of the analyses should be explained more explicitly.

*The introduction is expanded to explain this better.*

*"A way to avoid the discussion on which emission scenario to use and which scenario is the most likely — a discussion that is sometimes heated (Hausfather & Peters, 2020; Schwalm et al., 2020) — is to use global warming levels (GWL). Instead of a fixed period of time in a certain scenario GWLs focus on the period when a particular level of global warming is reached. For example, GWL2 is the period when +2°C global warming is reached compared to pre-industrial times. This period may occur at different times in different models – instead of consistency in*

*time between the members of the ensemble there is thus a consistency in the magnitude of temperature increase. In that way, using GWLs is a powerful method since it is possible to mix simulations using different scenarios to create larger ensembles; and since it reduces the uncertainty around the choice of emission scenarios (Maule et al., 2017). One example of how to use GWLs for regional data is found in Strandberg et al. (2024b). The mixing of emission scenarios in GWLs can nevertheless be criticised because the trends are different between scenarios (Bärring & Strandberg, 2018); a GWL based on RCP2.6 does not have the same characteristics as a GWL based on RCP8.5. This means that also a GWL ensemble is sensitive to how it is constructed with regards to which models and scenarios that are used as input.*

*Since it is likely that global warming will reach +2 °C within this century, and since the Paris Agreement speaks of a keeping the temperature rise to below 2 °C, it is natural that descriptions of projected climate change are formed around a 2 degree warmer world. The question is how such global warming levels are influenced by the climate models and emission scenarios used to calculate them."*

*Language and Clarity:*

The manuscript requires revision for language and writing style. Scientific clarity and fluency should be improved throughout. Several sections would benefit from more concise descriptions, while others lack necessary technical details—particularly regarding the ensemble setup and methodology.

*The manuscript has been rearranged, large parts have been rewritten and several figures are remade. We hope that this makes things more clear.*

*Tables:*

Table 1 – The asterisk (*) is unnecessary, as the information it provides is already evident from the table.

*While it true that it is possible to deduce which simulations that are used by looking at the scenario names in the table, we still think that adding asterisks guides the reader, making it easier to quickly find the right simulations.*

Table 2 – The selection of indices is likely relevant for the Scandinavian climate; however, presenting absolute changes in some indicators without referencing the baseline climate limits their interpretability. For example, many regions in Scandinavia experience few or no summer days, while others have around 30 in the reference period. Similarly, heavy precipitation of 10 mm/day is common along the Norwegian coast but rare in the polar regions of northern Sweden. Including percentile-based metrics or relative changes in indicators like Rx1day could improve clarity. Additionally, for consistency with other studies, replacing the Dry Days (dd) index with wet-day frequency and presenting its relative change may provide more meaningful interpretation.

*We added a new supplementary figure showing absolute values for the climatology in 1971-2000.*

*Figures:*

The figure requires major revision. Figures should be self-explanatory, featuring clear captions, meaningful legends, and clean visualization to convey the message without relying on the main text.

*We revised several figures and captions and hope that they are now acceptable.*

Figure 2 – As this paper aims to provide an overview of projected climate change in Scandinavia, including precipitation changes would be beneficial. Additionally, while the paper focuses on global warming levels, it presents end-of-century temperature changes for different emission scenarios, which is somewhat inconsistent with the emphasis on global warming levels expressed elsewhere in the text.

*Projected precipitation changes are shown in Fig. 3, and in supplementary figures S2-6. Figures 2-5 in the revised version are all based on time slices. We see a consistency in this. It is only the sub-ensemble analysis that is only based on GWLs.*

Figure 3 – Panels c, d, e, g, and h require representation of historical climatology to enhance interpretability (see also comment in the tables section).

*We added a new supplementary figure showing absolute values for the climatology in 1971-2000.*

Figure 5 and 6 – The content presented is unclear. The terms "South" and "North" are undefined, and the GCMs/RCMs corresponding to the y- and x-axis indices are not indicated. The meaning of "csu" is also unclear. Captions require substantial revision to improve clarity. Figures should be self-explanatory, with comprehensive captions and legends that convey the intended message without relying on the main text.

*These captions are extended to improve clarity.*

**Some specific comments**

L30 – The temperature response in Europe is correlated but warms at stronger rates.

*Changed as suggested.*

L33 – The paragraph on climate models needs to be introduced stating why climate models are employed.

*We added this to the beginning of the paragraph: "Climate models are our main tool to make projections of future climate change."*

L40 – best available –> most comprehensive

*Changed as suggested.*

L42 – they also allow for a probabilistic assessment of potential changes. In general, I would state the ability to employ a wider set of statistical tests at the end not as prominent.

*We changed the sentence to: "A key advantage of using climate model ensembles, like the CORDEX ensemble, is that they allow for a probabilistic assessment of potential changes, uncertainty estimations and a wider set of statistical tests."*

L46 – this sounds as if this paper would introduce the EURO-CORDEX data, however, it is rather the presentation of an analysis of a dynamically downscaled ensemble.

*We do actually present a new data set. The bias adjusted ensemble over Scandinavia has never been properly presented before. To make this more clear, we change the sentence to: "Here we present a new dynamically downscaled and bias adjusted ensemble of climate*

*projections for Sweden. Compared to the previous ensemble (Kjellström et al., 2016) improvements include ..."*

L58 – The "construction" section that follows is inconclusive and appears primarily driven by the choices of the EURO-CORDEX initiative and subdividing the ensemble by GCM, RCM and RCP, as clarified in later parts of the study.

*The Introduction has a completely new ending that hopefully makes this more clear.*

L75 – L80 (2.2 Bias adjustment) – Please add more detail on the set-up of the quantile-mapping implementation. This is particularly important as some quantile-mapping implementation preserve trends (climate change signals) while other can modify it. This is of high relevance for the present paper as you compare quantile-mapped RCM data with raw GCM data, which is generally not a proper and fair comparison (see also later).

*The chapter on bias adjustment is expanded to better describe the details of the Midas bias adjustment method.*

L180 – GMC à GCM

*Corrected.*

L183 – sentence seems to be incomplete.

*This sentence was removed in the revision.*

L202 – consistency in the rate of global warming.

*There is not necessarily a consistency in rate of global warming, as the temperature trend differs between models and scenarios. This depends on how the GWL ensemble is constructed. Nevertheless, we see the point in not writing "climate change" as this could mean several different things. Instead we changed to: "... instead of consistency in time between the members of the ensemble there is thus a consistency in the magnitude of temperature increase"*

L267 – warming since pre-industrial conditions. ¨

*We rephrased to this: "In Scandinavia, like most of Europe, the warming since pre-industrial time was about twice the global mean at the end of the 20th century"*

L274 – Out of interest, is there a possible explanation for (1) the warming being twice as large as the global average since pre-industrial times, and (2) the subsequent leveling off of this trend in the future (e.g., due to sea-ice dynamics, polar processes, or topography-related factors)?

> *1) The main reason is that land areas warm faster than oceans. This effect gets and extra boost in Scandinavia by the arctic amplification feed backs. When snow and ice retreats, the albedo decreases and more energy is absorbed by the surface. Furthermore, when sea ice disappears the heat from the relatively warm waters can more easy reach the atmosphere, thereby making it warmer.*
> *2) As far as we know this has not really been studied. A possible explanation is that the snow and ice feed backs loose their powers when the snow season gets too short.*

---

## Author Response (AR2)

Many thanks for all new comments and for an active editorship. Answers to the comments are given below.

**Reviewer #1**

2nd Review of 'Projected climate change in Fennoscandia — and its relation to ensemble spread and global trends' by Strandberg et al.

This is my second review of this manuscript. The new version of the manuscript is much improved, and the authors have sufficiently answered most of my concerns. I still think that the results of the bias correction should be discussed, especially since one important results is the comparison of spread between RCM and GCM (e.g. 4.2). How does the bias correction impact the spread in RCMs?

*We have now also included the raw RCM ensemble in the analysis. It is now included in the new figure 5. We also added text to describe it in section 3.4: "The CORDEX RCM ensemble is compared to its raw equivalent, where no bias adjustment has been performed, to assess the impact of bias adjustment on the climate change signal. The means and spreads are similar in both RCM ensembles, but the raw ensemble systematically shows smaller changes. Although small, these differences are significant in DJF, and in JJA under RCP8.5."*

*And discussed in 4.2: "In this study, bias-adjusted RCMs are compared to non-adjusted GCMs. Bias adjustment may reduce model spread in absolute values since systematic biases are minimised and all models are forced towards the reference data. Here, it systematically increases the climate change signal in the RCM ensemble. Although this increase is in many cases significant, it is relatively small, and the raw RCM ensemble is more similar to the bias-adjusted RCM ensemble than to any of the GCM ensembles. Consequently, the differences between GCMs and RCMs are likely not explained by the application of bias adjustment."*

Minor comments

Line 138-141 and 177-181 are duplicated.

*These lines are now merged under the new section 2.5 Definition of global warming levels*

Although the manuscript is well written, there is still need for a read trough and correct grammar.

**Reviewer #2**

I thank the authors for addressing my previous comments and acknowledge the extensive changes to the manuscript. I think that the changes in the methods section improved the accessibility of the paper greatly. Also, the reordering of the Results section makes the structuring of the results more logical. Lastly, I appreciate the addition of a discussion section considering my suggestions for content.

At this stage I only have a few minor comments.

P3 L75-77: "Since this RCM ..." You can remove this sentence as you are repeating this information in the next sentence.

*Done as suggested.*

P3 L90f: Can you maybe formulate the fourth topic a bit more straight to the point. For example: The role of climate model and emission scenario selection on the projected changes in temperature and precipitation at a GWL +2°C. This is important because ....

*This point is changed to:*

*"iv)      The role of climate model and emission scenario selection in projected changes in temperature and precipitation at +2°C global warming. This is particularly important because the Paris Agreement (UNFCCC, 2015) aims to keep temperature rise well below 2 °C. Consequently, descriptions of projected climate change naturally focus on a two-degree warmer world."*

P7 L137-141: "The GWLs are calculated ...". The same information is also presented in section 2.5 (P9 L177-184). I would suggest merging the information of these two sections and either place that information in one of the sections where this fits the best or create a new subsection, e.g. "Definition of global warming levels". I think the latter might be the best option.

*These lines are now merged under the new section 2.5 Definition of global warming levels*

**#3**

No further comments

**Editor:**

I agree with the reviewer that in order to thoroughly assess the effect of bias adjustment on the climate change signal, it is desirable to assess the spread and signal in GCM, raw RCM and bias adjusted RCM. The intermediate step of assessing the signal in the raw RCM allows disentangling the effect of downscaling from the effect of bias adjusting.

Given the small and varying size of the ensembles the min-max range is not a meaningful metric for an uncertainty range as it is extremely sensitive to the exact ensemble configuration. It is definitely better to rely on variance or interquantile range in this respect. That said, there are some outliers in Fig. 5 that deserve more thorough discussion. I am looking forward to receive a revised version of the manuscript along with a point by point response to the remaining reviewer concerns.

*We have now also included the raw RCM ensemble in the analysis, see comment above. Figure 5 does not only show min and max values but also the range between the 10th and 90th percentiles, which we think is an acceptable way to show the spread.*